# Structure and mechanism of biosynthesis of *Streptococcus mutans* cell wall polysaccharide

Jeffrey S. Rush[1], Svetlana Zamakhaeva [2], Nicholas R. Murner[2], Pan Deng[3,7], Andrew J. Morris [4,8], Cameron W. Kenner[2], Ian Black[5], Christian Heiss[5], Parastoo Azadi[5], Konstantin V. Korotkov [1], Göran Widmalm [6] & Natalia Korotkova [1,2] ✉

*Streptococcus mutans*, the causative agent of human dental caries, expresses a cell wall attached Serotype *c*-specific Carbohydrate (SCC) that is critical for cell viability. SCC consists of a polyrhamnose backbone of →3)α-Rha(1 → 2)α-Rha(1→ repeats with glucose (Glc) side-chains and glycerol phosphate (GroP) decorations. This study reveals that SCC has one predominant and two more minor Glc modifications. The predominant Glc modification, α-Glc, attached to position 2 of 3-rhamnose, is installed by SccN and SccM glycosyltransferases and is the site of the GroP addition. The minor Glc modifications are β-Glc linked to position 4 of 3-rhamnose installed by SccP and SccQ glycosyltransferases, and α-Glc attached to position 4 of 2-rhamnose installed by SccN working in tandem with an unknown enzyme. Both the major and the minor β-Glc modifications control bacterial morphology, but only the GroP and major Glc modifications are critical for biofilm formation.

The Gram-positive bacterium *Streptococcus mutans* is a normal inhabitant of the human oral cavity recognized as a major etiological agent of human dental caries[1]. Ability of this organism to colonize tooth surfaces forming biofilms is directly associated with the development of dental caries[2]. Similar to other streptococcal species, *S. mutans* strains decorate peptidoglycan with rhamnose (Rha)-containing cell wall polysaccharides, so called rhamnopolysaccharides, that are important for the viability of these bacteria[3–6]. Streptococcal rhamnopolysaccharides are functional homologs of teichoic acid glycopolymers that are present in the cell walls of many Gram-positive bacteria[7–9]. Loss of cell wall polysaccharides in *S. mutans* is associated with a pleiotropic phenotype that includes abnormal cell morphology, enhanced cellular autolysis and defective biofilm formation[5,10,11].

Streptococcal rhamnopolysaccharides have attracted significant attention as promising candidates for the development of glycoconjugate vaccines[12–19] and as receptors for bacteriophage-encoded endolysins[20,21], a novel class of antimicrobials that kill bacteria by degrading peptidoglycan[22,23]. Furthermore, enzymatic steps in rhamnopolysaccharide synthesis are considered to be novel targets for antibacterial drug design[24,25].

*S. mutans* strains are classified into four serotypes, *c, e, f and k*, based on variations in the molecular structures of their rhamnopolysaccharides[26]. Epidemiological surveys indicate that 70–80% of strains found in the oral cavity are classified as serotype *c*, followed by serotypes *e* (about 20%) and *f* and *k* (less than 5% each)[26,27]. The major structural feature of all *S. mutans*

[1]Department of Molecular and Cellular Biochemistry, University of Kentucky, Lexington, KY, USA. [2]Department of Microbiology, Immunology and Molecular Genetics, University of Kentucky, Lexington, KY, USA. [3]Department of Pharmaceutical Sciences, College of Pharmacy, University of Kentucky, Lexington, KY, USA. [4]Division of Cardiovascular Medicine and the Gill Heart Institute, University of Kentucky, Lexington, KY, USA. [5]Complex Carbohydrate Research Center, University of Georgia, Athens, GA, USA. [6]Department of Organic Chemistry, Arrhenius Laboratory, Stockholm University, Stockholm, Sweden. [7]Present address: Jiangsu Key Laboratory of Neuropsychiatric Diseases and College of Pharmaceutical Sciences, Soochow University, Suzhou, Jiangsu, China. [8]Present address: Department of Pharmacology and Toxicology, University of Arkansas for Medical Science and Central Arkansas Veterans Affairs Healthcare System, Little Rock, AR, USA. ✉e-mail: nkorotkova@uky.edu

rhamnopolysaccharides is a linear backbone structure, composed of repeating →3)α-Rha(1→2)α-Rha(1→ disaccharides. In *S. mutans* serotype *c*, the polysaccharide, called the serotype *c*-specific carbohydrate or SCC (also referred to elsewhere as rhamnose-glucose polysaccharide or RGP[28]), is reported to contain α-glucose (Glc) side-chains attached to the 2-position of 3-linked Rha units[14]. Alternatively, serotype *e* has β-Glc attached to the corresponding hydroxyl group of Rha[29]. Serotype *f* contains α-Glc or (Glc)₂ side-chain attached to the 3-position of 2-linked Rha and serotype *k* has α-galactose side-chains attached to the corresponding position on Rha[14]. In *S. pyogenes*, the homologous polysaccharide called the Group A Carbohydrate (GAC) displays a similar core polyrhamnose backbone structure modified with *N*-acetyl-β-glucosamine (GlcNAc) side-chains[30]. Our recent detailed compositional and structural analysis of GAC revealed that approximately 25% of the GlcNAc side-chains are decorated with glycerol phosphate (GroP) moieties at the C6 hydroxyl group[8]. This modification had gone unnoticed in the past decades, presumably due to loss of GroP during preparation. The presence of GroP was also detected on the Glc side-chains of *S. mutans* SCC, although the exact structure of the GroP-Glc linkage has not yet been reported[8,9].

This study was undertaken to investigate SCC structure and poorly understood steps of SCC biosynthesis. Most genes responsible for SCC biosynthesis in *S. mutans* are arranged in a single cluster that can be functionally divided into two regions (Fig. 1A). Region 1 is conserved among all *S. mutans* serotypes and many streptococcal species and encodes the proteins for polyrhamnose biosynthesis and export[3,13,31]. Region 2 is species and serotype-specific and encodes proteins for side-chain attachment and GroP modification[8,13,31]. Assembly of the polyrhamnose backbone is initiated on the cytoplasmic side of the plasma membrane by the transfer of GlcNAc-phosphate from UDP-GlcNAc to a carrier lipid, undecaprenyl-phosphate (Und-P), catalyzed by the TarO/TagO/WecA homolog[32], known as RgpG in *S. mutans*[33], followed by sequential addition of L-Rha units catalyzed by SccB, SccC and SccG[34,35]. The lipid-linked polyrhamnose intermediate is then transported into the periplasm by an ABC transporter[35] where it is further decorated with the side-chains and GroP[8,9] (Fig. 1B).

The molecular details of Glc side-chain addition to SCC have not yet been described in detail. We previously reported that both SccN and SccP participate in decoration of SCC with the Glc side-chains, although SccN is required for the addition of the majority of the Glc moieties[9]. Moreover, only the Glc side-chains provided by SccN are recognized by the GroP transferase, SccH, for the attachment of GroP moieties[9]. The Glc side-chains and GroP decorations are critically important in the biology and pathogenesis of *S. mutans*. The SccN and SccH-deficient mutants exhibit self-aggregation, aberrant cell morphology and increased autolysis[9]. These defects are due to mislocalization of the cell division proteins in these mutants. Furthermore, the SccN-deficient mutant shows a defect in biofilm formation and exopolysaccharide production, and attenuated virulence in a rat model of dental caries[36].

Here, we show that SccN and SccP are glucosyl phosphoryl undecaprenol (Glc-P-Und) synthases that catalyze the formation of β-Glc-P-Und and α-Glc-P-Und, respectively (Fig. 1B). SccM and SccQ are Glc-P-Und:polyrhamnose glucosyltranferases that transfer Glc from Glc-P-Und to the polyrhamnose chain to form the Glc side-chains (Fig. 1B). Glycosyl linkage analysis together with NMR analysis indicate that SccN and SccM are responsible for the formation of the most abundant α-Glc modification, whereas SccP and SccQ contribute to the formation of the minor β-Glc modification. Furthermore, we report that the GroP decoration is critically important for *S. mutans* biofilm formation. Our NMR studies reveal that GroP is attached exclusively on the 6-OH of the α-Glc side-chains donated by SccN and SccM.

## Results

### SccN and SccP are UDP-Glc:Und-P glucosyl transferases

In bacteria, multi-component transmembrane glycosylation complexes catalyze the attachment of sugars to a variety of glycans. This machinery requires a GT-A fold glycosyltransferase to synthesize an Und-P linked glycosyl donor, a flippase to move the lipid intermediate across the cell membrane and a periplasmically-oriented GT-C fold glycosyltransferase to transfer the sugar from the lipid intermediate to the nascent polysaccharide[37,38]. In *S. mutans* serotype *c*, Region 2 encodes two GT-A fold glycosyltransferases SccN and SccP, and two GT-C fold glycosyltransferases SccM and SccQ (Fig. 1A)[9]. We hypothesized that the Glc side-chains of SCC in *S. mutans* are donated to the polyrhamnose backbone from an undecaprenol-linked intermediate, Glc-P-Und, as has been reported for an analogous biosynthetic pathway in *S. pyogenes*[38]. To test this idea, SccN and SccP were expressed on a plasmid in the *E. coli* JW2347 strain, which is Glc-P-Und synthase deficient. In membrane fractions prepared from the recombinant strains, both synthases actively catalyzed the formation of a [³H]glucolipid when incubated in vitro with UDP-[³H]Glc and Und-P. In contrast, the activity was negligible in membrane fractions of *E. coli* JW2347 (Supplementary Table 1). A kinetic analysis of *E. coli* expressed SccN and SccP revealed that both enzymes show a high affinity for UDP-Glc and Und-P, whereas SccP possesses a slightly lower affinity for UDP-Glc, but higher affinity for Und-P than SccN. However, the apparent maximal rates for SccN are higher than that of SccP (Fig. 2A and Table 1). These data confirm that SccN and SccP are both Glc-P-Und synthases. To understand if SccN and SccP are active in *S. mutans*, the membrane fractions prepared from *S. mutans* Xc wild type (WT), the *sccN* and *sccP* deletion mutants (Δ*sccN* and Δ*sccP*), and the double deletion mutant Δ*sccN*Δ*sccP* were incubated with UDP-[³H]Glc and Und-P, in vitro. Preliminary experiments revealed that *S. mutans* membrane fractions synthesized two classes of glucolipids: a major glucolipid product with chromatographic properties (thin layer silica gel G, see Methods) similar to [³H]Glc-P-Und formed in the in vitro reactions with *E. coli* expressed SccN and SccP, described above, and an additional minor glucolipid which is most likely a glucosyldiglyceride. Subsequently, to eliminate the glycerolipid contaminant, organic extracts of in vitro reactions were subjected to mild alkaline de-acylation prior to analysis for [³H]-Glc-P-Und. The synthase activity in the Δ*sccN* membranes was greatly reduced compared to WT cells (Supplementary Fig. 1a and Fig. 2B). In contrast, deletion of SccP had a modest effect on Glc-P-Und synthase activity. Membrane fractions from Δ*sccN*Δ*sccP* did not catalyze the synthesis of Glc-P-Und. The [³H] glucolipid products formed in the reactions containing the Δ*sccN* and Δ*sccP* membranes were not distinguishable by thin layer chromatography (TLC) to that formed in the WT reactions (Supplementary Fig. 1b and Fig. 2B). The activities were stimulated by the exogenous addition of Und-P as a suspension in CHAPS detergent (Supplementary Fig. 1a), and inhibited by amphomycin which is known to form an insoluble complex with Und-P (Supplementary Fig. 1c, d)[39]. Further analysis showed that the [³H]glucolipids are anionic, sensitive to mild acid and resistant to mild alkali as expected for a glycosyl phosphoryl isoprenoid. In addition, a co-migrating compound, purified from the *S. mutans* membrane fraction by organic solvent extraction and preparative TLC, yielded a molecular ion, $m/z = 1007.7$ by ESI-MS which is expected for Glc-P-Und (Supplementary Fig. 1e). Thus, our data indicate that both SccN and SccP catalyze the synthesis of Glc-P-Und.

### SccN and SccP synthesize β-Glc-P-Und and α-Glc-P-Und, respectively

Although the enzymatic products formed by SccN and SccP from UDP-Glc and Und-P were chromatographically indistinguishable, and both glycosyltransferases are active in *S. mutans* membranes, only the Glc side-chains provided by SccN are decorated with GroP[9]. GT-A type glycosyltransferases are known to be readily reversible in the presence

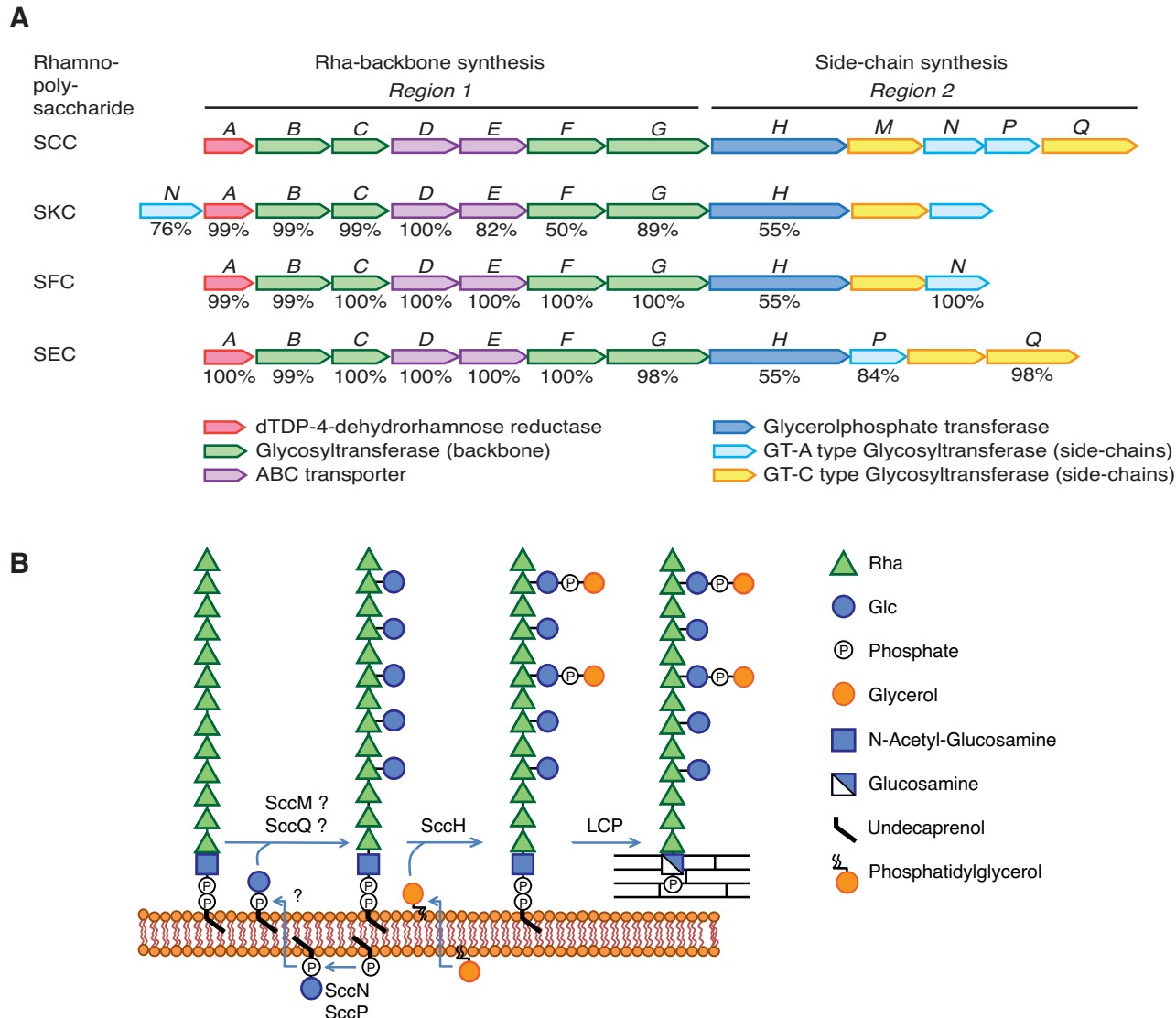

**Fig. 1 | Schematic representation of the *S. mutans* rhamnopolysaccharide biosynthetic gene clusters and the proposed mechanism of SCC modification with the Glc side-chains and GroP. A** Schematic representation of the genetic loci encoding enzymes involved in the biosynthesis of rhamnopolysaccharides expressed by *S. mutans* serotypes *c*, *k*, *f*, and *e*: Serotype *c* Carbohydrate (SCC), Serotype *k* Carbohydrate (SKS), Serotype *f* Carbohydrate (SFC) and Serotype *e* Carbohydrate (SEC), respectively. Genes in Region 1 participate in the synthesis of the polyrhamnose backbone. Genes in Region 2 participate in modification of the polyrhamnose backbone with the side-chains and GroP. The SCC gene cluster contains genes designated *sccABCDEFGHMNPQ* corresponding to smu.824-835 in *S. mutans* UA159 (GenBank: AE014133.2). The SKC gene cluster represents the SMULJ23_1190-SMULJ23_1180 gene locus in *S. mutans* LJ23 (GenBank: AP012336.1). The SFC gene cluster represents the K2F51_02855-K2F51_02900 gene locus in *S. mutans* OMZ175 (GenBank: CP082153.1). The SEC gene cluster represents the CO204_06335-CO204_06385 gene locus in *S. mutans* LAR01 (GenBank: CP023477.1). The percentage of amino acid identity to proteins encoded by the SCC gene cluster are noted below each gene. **B** Model illustrating SCC side-chain assembly and transfer to peptidoglycan in *S. mutans* serotype *c*. Side-chain addition starts with the synthesis of Glc-P-Und, on the cytosolic surface of the plasma membrane, catalyzed by two independent Glc-P-Und synthases (SccN and SccP) using UDP-Glc and Und-P as substrates. Following translocation of Glc-P-Und to the periplasmic surface (mediated by currently unidentified flippase proteins, indicated as a question mark), the GT-C type transferases, SccM and SccQ, transfer Glc from Glc-P-Und to the polyrhamnose backbone at the 2- and 4-positions of 3-Rha components, respectively. In addition, transfer of Glc from Glc-P-Und to the 4-position of 2-Rha takes place, but only to a limited extent (glycosyltransferase unknown). Following glucosylation, SccH transfers GroP from phosphatidylglycerol to the 6-OH of Glc residues found only at the 2-position of 3-Rha. After side-chain assembly has been completed, SCC is transferred to peptidoglycan by LytR-Cps2A-Psr (LCP) family proteins.

of the appropriate nucleoside diphosphate, re-forming the nucleoside-diphospho-sugar and releasing Und-P[40,41]. To investigate if the enzymatic products of SccN and SccP are interchangeably equivalent substrates in the reverse direction to reform UDP-Glc, we purified the [3H] Glc-P-Und products by preparative TLC and tested them for reverse synthesis by incubation with UDP in the presence of each of the enzymes. Interestingly, the enzymatic product of the SccN Glc-P-Und synthase reaction is efficiently discharged in a time-dependent reaction by SccN, but not by SccP (Fig. 2C and Supplementary Fig. 2). The enzymatic product formed by SccP is likewise discharged by SccP, but not by SccN (Fig. 2C and Supplementary Fig. 2). This experiment indicates that the Glc-P-undecaprenols synthesized by SccN and SccP are most likely unique stereoisomers.

To identify the anomeric configuration of the Glc-P-Und lipids produced by *S. mutans*, we extracted preparative scale quantities of Glc-P-Und from WT and the Δ*sccN*, Δ*sccP*, Δ*sccM* and Δ*sccQ* mutants, purified by multiple rounds of preparative TLC and analyzed by ESI-MS liquid chromatography. The mass spectra obtained by ESI-MS from

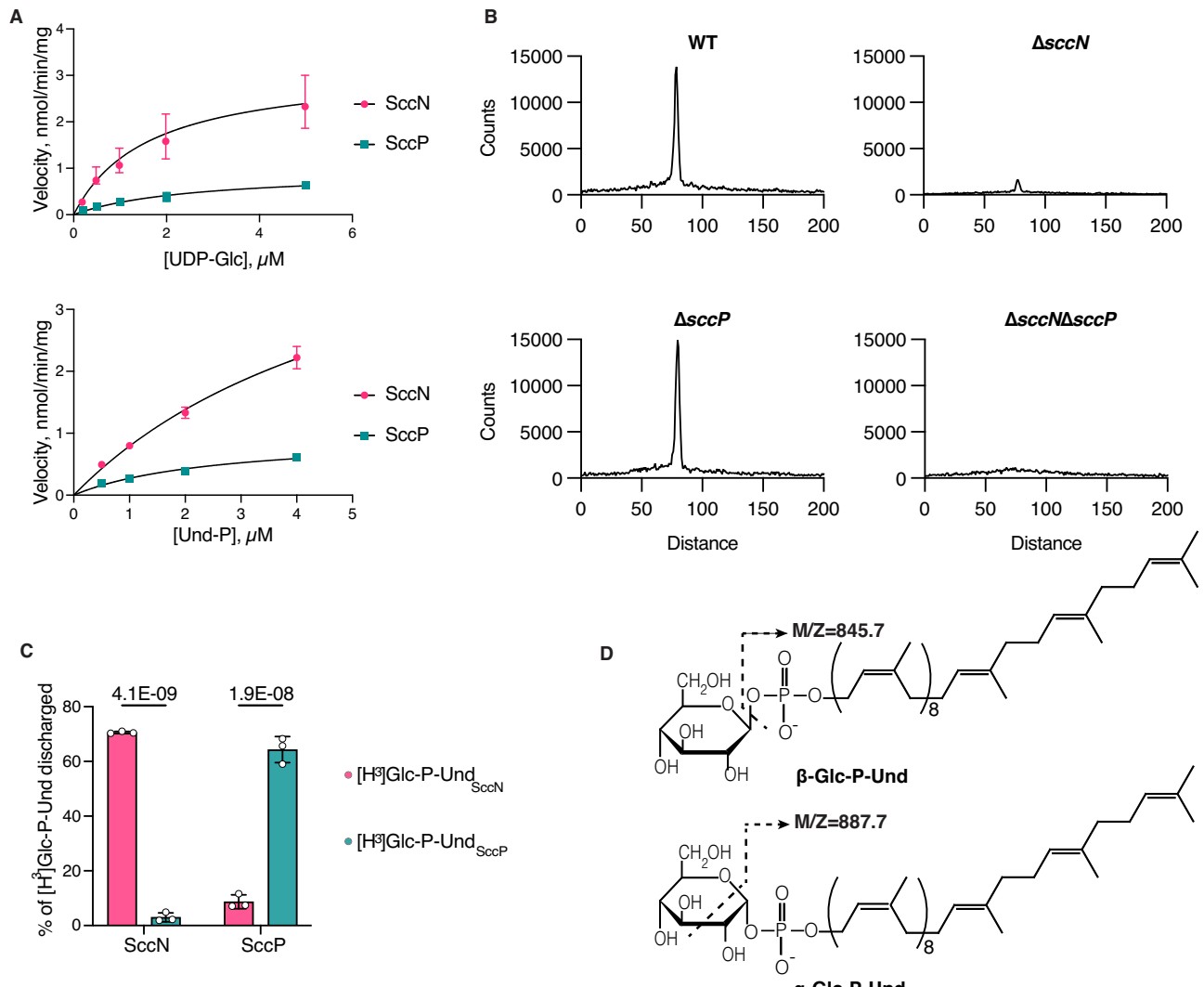

**Fig. 2 | SccN and SccP synthesize alternative stereoisomers of Glc-P-Und.**
**A** Kinetic analysis of Glc-lipid synthases in membrane fractions from the *E. coli* JW2347 strain expressing *sccN* or *sccP* on a plasmid. Membrane fractions were assayed for the formation of [³H]Glc-lipids in the presence of increasing amounts of UDP-[³H]Glc or Und-P (undecaprenol phosphate), added as a dispersion in 1% CHAPS. Median average velocity and standard errors for three separate experiments are plotted versus substrate concentrations. Apparent kinetic parameters for UDP-Glc and Und-P were calculated using GraphPad Prism 9.3 software, Michaelis-Menten enzyme kinetics tool. Km and Vmax for UDP-Glc and Und-P are presented in Table 1. The data are derived from three separate experiments. **B** Thin layer chromatography (TLC) of alkali-stable [³H]Glc-lipids from in vitro incubations of the membranes of *S. mutans* mutants with UDP-[³H]Glc. Membrane fractions from *S. mutans* WT, Δ*sccN*, Δ*sccP*, or Δ*sccN*Δ*sccP* were incubated with UDP-[³H]Glc and analyzed for [³H]Glc-lipid synthesis by TLC following mild-alkaline methanolysis as described in Methods. [³H]Glc-lipids were detected by scanning with an AR2000 Bioscan Radiochromatoscanner. The results are representative of three

independent experiments. **C** Reverse reactions of SccN and SccP to reform UDP-Glc from their respective enzymatic products following incubation (37 °C, 40 min) with UDP. [³H]Glc-P-Und$_{SccN}$, synthesized by SccN, and [³H]Glc-P-Und$_{SccP}$, synthesized by SccP, were tested as substrates in the discharge reactions, containing solubilized, partially-purified SccN or SccP, as described in Methods. Columns and error bars represent the mean and S.D., respectively (three biologically independent replicates). *P* values were calculated by two-way ANOVA with Šídák's multiple comparison test. **D** Origin of diagnostic ion fragments used to deduce the anomeric configuration of Glc-P-Und. If the stereochemistries of the phosphodiester at C1 of glucose and the 2-OH are configured *trans* (β-anomeric configuration) then [Und-HPO$_4$]⁻ (*m/z* = 845.7) is the primary high molecular weight ion detected in the mass spectrum. However, if the stereochemistries of glucose at C1 and the 2-OH group are configured *cis* (α-anomeric configuration) then an additional ion fragment with the composition of [Und-PO$_4$-C$_2$H$_3$O]⁻ (*m/z* = 887.7), formed by 'cross-ring' fragmentation as shown, is found[42]. Source data for **a** and **c** are provided as a Source Data file.

each of the purified Glc-P-Und isolates contained high mass ions derived from [M-H]⁻ (*m/z* = 1007.7), [M-H$_2$O-H]⁻ (*m/z* = 988.7), [Und-PO$_4$-C$_2$H$_3$O]⁻ (*m/z* = 887.7), [Und-HPO$_4$]⁻ (*m/z* = 845.7) and [glucosyl-1-PO$_3$]⁻ (*m/z* = 241.7), as expected (Supplementary Fig. 3). During fragmentation of sugar 1-phosphates by ESI-MS, if the phosphodiester at C1 and the hydroxyl group at C2 are oriented *trans* to the sugar ring (i.e., the functional groups reside on opposite sides of the plane of the sugar ring) then [Und-HPO$_4$]⁻ (*m/z* = 845.7) is released as the highest molecular weight fragment (Fig. 2D). However, if the relative orientation of the groups at C1 and C2 is arranged *cis* (i.e., the functional groups

reside on the same side of the plane of the sugar ring), then an additional cross-ring fragmentation product, [Und-PO$_4$-C$_2$H$_3$O]⁻ (*m/z* = 887.7) is released (Fig. 2D). Wolucka, et al. have reported that the ratio of the relative abundance of the [Und-PO$_4$-C$_2$H$_3$O]⁻ ion to the relative abundance of the [Und-HPO$_4$]⁻ ion is close to one for sugar 1-phosphates with C1-O and C2-OH in *cis* configuration, but close to zero for sugar 1-phosphates with a *trans* orientation of these hydroxyl groups[42]. Importantly, the spectra derived from the Δ*sccN* glucolipid compared with the spectra of the Δ*sccP* glucolipid revealed clear differences in the relative abundances of the fragment ions arising from

loss of $[Und-HPO_4]^-$ ($m/z = 845.7$) and the cross-ring fragmentation pathway ($m/z = 887.7$). Table 2 shows that the ratio of the $[Und-PO_4-C_2H_3O]^-$ ($m/z = 887.7$) fragment to $[Und-HPO_4]^-$ ($m/z = 845.7$) is approximately 0.14 in the Glc-P-Und isolated from the $\Delta sccP$ mutant (indicating that the C1- and C2-groups are in *trans* orientation), whereas this ratio is approximately 1.02 in the Glc-P-Und isolated from the $\Delta sccN$ mutant (C1- and C2-groups oriented *cis*). This result is consistent with the conclusion that the $\Delta sccN$ strain (expressing SccP) synthesizes an α-Glc-P-Und and that $\Delta sccP$ (expressing SccN) synthesizes a β-Glc-P-Und. The WT strain shows a relative abundance ratio of $[Und-PO_4-C_2H_3O]^-$ to $[Und-HPO_4]^-$ of 0.49, indicating that a mixture of the two stereoisomers is most likely present. Furthermore, it is reassuring that the spectra of the glucolipids from the $\Delta sccM$ and $\Delta sccQ$ mutants, in which SccN and SccP expression is unaffected, give relative ion abundance ratios that are similar to that of the WT strain.

## The functions of SccN, SccP, SccM and SccQ in SCC biosynthesis

To establish the exact roles of the SccN, SccP, SccM and SccQ glycosyltransferases in SCC glucosylation, we purified sodium borohydride reduced SCCs from WT, $\Delta sccN$, $\Delta sccP$, $\Delta sccM$ and $\Delta sccQ$ and the double deletion mutant $\Delta sccM\Delta sccQ$. SCC variants were analyzed for glycosyl composition as TMS-methyl glycosides following methanolysis. The analysis found primarily Rha and various amounts of Glc, along with *N*-acetyl-D-glucosaminitol (GlcNAcitol), derived from the reducing end GlcNAc of SCC[43], and lesser amounts of non-reducing end GlcNAc (Supplementary Fig. 4). Deletion of *sccN* reduces (~80%) the Glc content of SCC, whereas deletion of *sccP* has a relatively minor affect, in agreement with our previous study[9]. It is worth noting that the reduction in Glc content observed in the $\Delta sccP$ SCC, is approximately the same as the residual Glc content found in the $\Delta sccN$ SCC, and, as we have show previously, the Glc content in the $\Delta sccN\Delta sccP$ SCC is reduced even further[9]. The Glc contents of the SCCs isolated from $\Delta sccM$ and $\Delta sccQ$ are each reduced to approximately 1/3rd of the WT content. Surprisingly, the Glc content of the SCC isolated from the mutant lacking both *sccM* and *sccQ* is about half of the WT content, suggesting that *S. mutans* has additional unknown glycosyltransferases responsible for the transfer Glc to the polyrhamnose backbone (Supplementary Fig. 4). As we explain below in detail, this increased Glc content in the double mutant in comparison to the single mutants is likely due to a pleiotropic effect of a loss of both SccM and SccQ on cell wall biosynthesis pathways.

To determine which Glc-P-Und stereoisomers are utilized by SccM and SccQ for the transfer of Glc to SCC and to identify the specific glycosidic products produced by these enzymes, the SCCs were subjected to glycosyl linkage analysis as partially methylated alditol acetates (PMAA). As expected from the glycosyl composition analysis (see above), linkage analysis identified Glc as the major non-reducing terminal sugar with minor amounts of non-reducing terminal Rha, GlcNAc and 4-substituted GlcNAcitol. The PMAA arising from 2-substituted Rha was the main component in all samples, but the quantity of 3-linked Rha was highly variable between the samples. Specifically, we found a major reduction in 2,3-disubstituted Rha and a concomitant increase in 3-linked Rha in the $\Delta sccN$ and $\Delta sccM$ SCCs (Fig. 3A, C and Supplementary Table 2). This is consistent with the function of these proteins in the transfer of Glc to the 2-OH of 3-substituted Rha. Additionally, the $\Delta sccQ$ SCC demonstrated a reduction (~50%) in the amount of 2,3-Rha. We attribute this reduction to a pleiotropic effect caused by the sequestration of Und-P into a metabolic intermediate rendered a 'dead-end' by the deletion of SccQ. The Und-P in this 'dead-end' product cannot be re-cycled and consequently accumulates in the cell, reducing the cellular Und-P available for polyrhamnose synthesis and SCC glucosylation. Surprisingly, a minor amount of 2,4-, 3,4- and 2,3,4-branched Rha was detected in some samples, suggesting that *S. mutans* possesses previously unreported 4-glucosyltransferase activities that can transfer Glc to either 2-linked, 3-linked, or 2,3-disubstituted Rha (Fig. 3A–E, and Supplementary Table 2).

Consistent with the glycosyl composition of the $\Delta sccN$ SCC, linkage analysis revealed a small residual amount of terminal Glc in this polysaccharide (Fig. 3F, Supplementary Table 2), which correlates very closely with the amount of 3,4-branched Rha. It is important to note that this Glc addition to the 4-position does not require SccN. The 2,3- and 2,3,4-substituted Rha branches are nearly absent in $\Delta sccN$ suggesting that the Glc addition to the 2-position of these branches requires SccN. In contrast, the 3,4-Rha branch is significantly enhanced when *sccN* is deleted (Fig. 3A, Supplementary Table 2). Hence, these observations suggest that the Glc attached to the 4-OH of 3,4-Rha is derived from the SccP pathway and the Glc attached to the 2-OH of the 2,3,4-branched Rha originates from the SccN product.

The signal for 3,4-disubstituted Rha is also increased in the $\Delta sccM$ SCC, at the expense of the 2,3,4-branched Rha component, and nearly absent in the $\Delta sccQ$ SCC, confirming the role of SccM in glucosylation of the 2-OH and strongly suggesting that the formation of the branch at the 4-position requires SccQ in addition to SccP (Fig. 3C, Supplementary Table 2). It is important to note that there is a very small signal for 3,4-Rha in the WT SCC. The increase of this branch in the $\Delta sccN$ and

## Table 1 | Kinetic parameters of Glc-P-Und synthases in membrane fractions from *E. coli* JW2347 expressing *sccN* or *sccP* on a plasmid

| Enzyme | SccN | | SccP | |
|---|---|---|---|---|
| *Substrate* | *Und-P* | *UDP-Glc* | *Und-P* | *UDP-Glc* |
| $Km$, app (μM) | 5.85 | 1.61 | 2.56 | 2.50 |
| Vmax, app (μmol·min$^{-1}$·mg$^{-1}$) | 5.42 | 3.16 | 0.96 | 0.93 |

## Table 2 | Relative abundances of fragment ions obtained by ESI MS/MS from Glc-P-Unds purified from various strains of *S. mutans* expressed as a percentage of the [M-H]$^-$ ion[a]

| *S. mutans* strains | Active Glc-P-Und Synthase | Relative abundance of ions (% [M-H]$^-$) | | Ratio | Glucolipid |
|---|---|---|---|---|---|
| | | $[Und-PO_4-C_2H_3O]^-$ *A* | $[Und-HPO_4]^-$ *B* | *A/B* | |
| WT | SccN/SccP | 40.2 | 82 | 0.49 | α/β-Glc-P-Und |
| $\Delta sccN$ | SccP | 65.4 | 64.2 | 1.02 | α-Glc-P-Und |
| $\Delta sccP$ | SccN | 12.7 | 89.4 | 0.14 | β-Glc-P-Und |
| $\Delta sccQ$ | SccN/SccP | 38.1 | 89.5 | 0.43 | α/β-Glc-P-Und |
| $\Delta sccM$ | SccN/SccP | 42.1 | 94.1 | 0.45 | α/β-Glc-P-Und |

[a]Total lipids from the various strains ($n = 5$) were extracted with chloroform/methanol, deacylated in KOH/methanol, purified by preparative TLC, and analyzed, concurrently, by Q-Exactive Orbitrap LC/MS under conditions described in Methods. Relative abundances of fragment ions are expressed as a percentage of the pertinent ion fragment relative to the parent ion ([M-H]$^-$). The observed $m/z$ value for [M-H]$^-$ was 1007.709, $[Und-HPO_4]^-$ was 845.657 and $[Und-PO_4-C_2H_3O]^-$ was 887.664. Source data are provided as a Source data file.

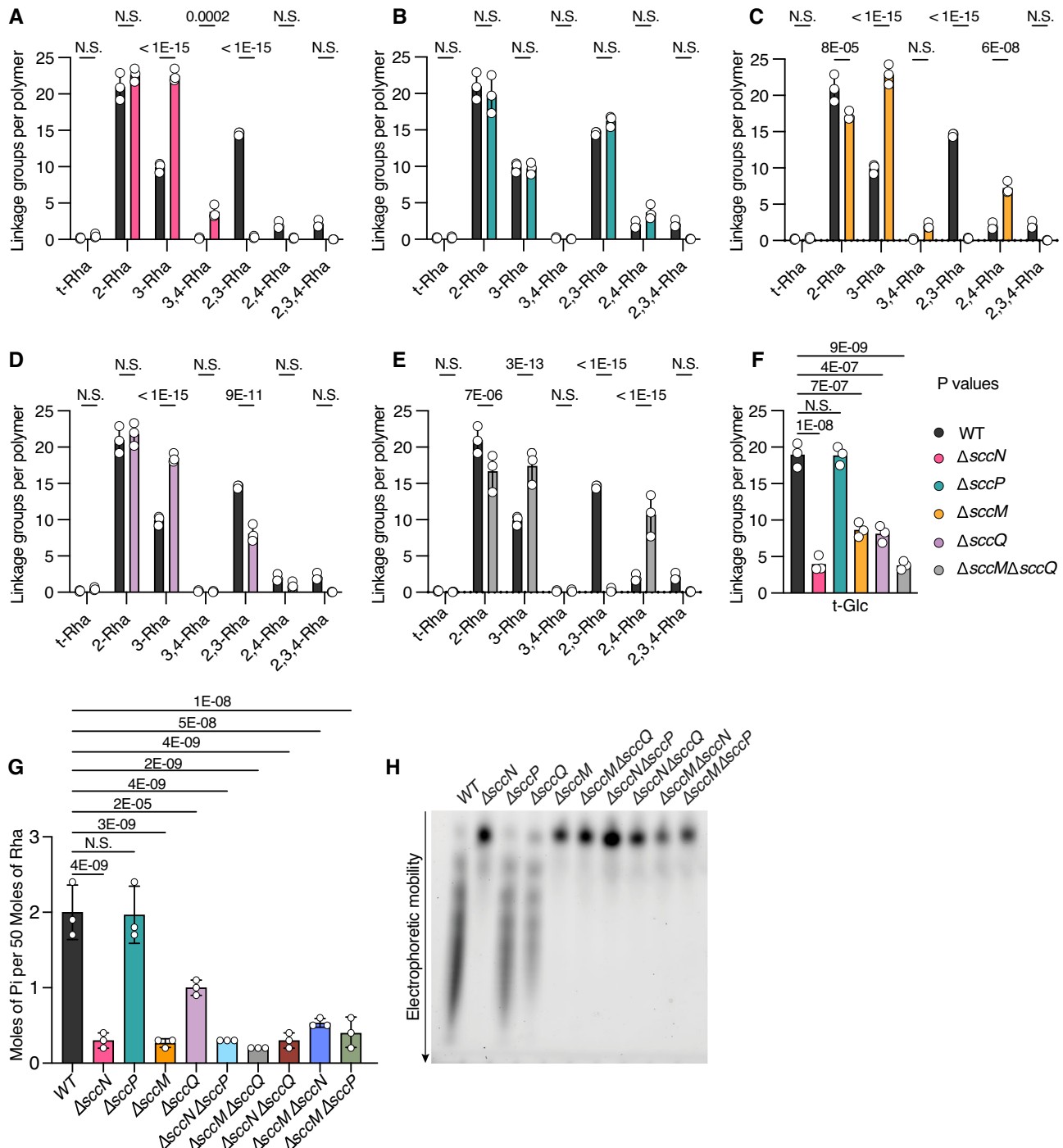

**Fig. 3 | Analysis of SCCs purified from *S. mutans* strains.** Quantitative comparison of the linkage groups in WT SCC with (**A**) Δ*sccN* SCC, (**B**) Δ*sccP* SCC, (**C**) Δ*sccM* SCC and (**D**) Δ*sccQ* SCC and (**E**) Δ*sccM*Δ*sccQ* SCC. **F** Recovery of non-reducing terminal Glc in SCCs purified from *S. mutans* strains. For this illustration, data are expressed as number of molecules normalized to a total of 50 rhamnose (Rha) per polymer (the proposed average size of the polyrhamnose backbone). The raw data for **A**–**F** are presented in Supplementary Table 2. **G** Phosphate content of SCCs isolated from *S. mutans* strains. Phosphate concentration was determined by the malachite green method following digestion with perchloric acid. Rha content was determined by a modified anthrone assay, as described in Methods. Data are expressed as nmol of phosphate in the sample normalized to 50 nmol Rha per molecule of SCC. In **A**–**G**, columns and error bars represent the mean and S.D., respectively ($n = 3$ independent replicates). *P* values were calculated by two-way ANOVA with Dunnett's multiple comparison test. Data used for plots shown in **A**–**F** are presented in Supplementary Table 2. **H** Fluorescence imaging analysis of SDS-PAGE of ANDS-labeled SCCs prepared from the indicated *S. mutans* strains. Representative image from at least three independent experiments is shown. Source data for **g** and **h** are provided as a Source data file.

Δ*sccM* SCCs may be a consequence of the deletion of *sccN* and *sccM*, respectively, allowing enhanced expression of the SccP/SccQ pathway. Lastly, of interest is the effect of these genetic deletions on the abundance of the 2,4-Rha branch. As we will discuss below, high resolution NMR analysis indicates that this side-branch is highly unusual in that it is formed by addition of Glc to the 4-position of 2-Rha, unlike the other branching residues. The 2,4-Rha signal is significantly reduced in Δ*sccN* and increased in Δ*sccP*, suggesting that the SccN product is required for this modification (Fig. 3A, B, Supplementary Table 2). In addition, 2,4-Rha content increases in the Δ*sccM*

SCC (probably due to increased availability of the SccN Glc-P-Und product), and does not change significantly in the ΔsccQ SCC (Fig. 3C, D, Supplementary Table 2). Furthermore, the 2,4-Rha content increases even further in SCC prepared from ΔsccMΔsccQ, confirming that neither of these GT-C glycosyltransferases are required to glucosylate the 4-OH of 2-Rha (Fig. 3E, Supplementary Table 2). Taken together, our data indicate that SccN and SccM are the major enzyme pair that mediate glucosylation of the 2-OH of the 3-substituted Rha. Furthermore, SccQ may utilize the SccP product, α-Glc-P-Und, to transfer Glc to the 4-position of 3-substituted and 2,3-disubstituted Rha to form the 3,4- and 2,3,4-Rha branches. It appears that the product of the SccN-mediated reaction, β-Glc-P-Und, is utilized by an unknown glycosyltransferase to transfer Glc to the 4-position of 2-Rha to form the 2,4-branched Rha.

### The Glc residues installed by SccN-SccM pair serve as GroP acceptors

To understand the roles of SccM and SccQ in the formation of the GroP acceptor side-chains of SCC, we compared the phosphate content of purified SCCs isolated from WT, ΔsccM, ΔsccQ, ΔsccN and ΔsccP, and five double mutants, ΔsccNΔsccP, ΔsccMΔsccQ, ΔsccNΔsccQ, ΔsccMΔsccN and ΔsccMΔsccP. The ΔsccN, ΔsccNΔsccP, ΔsccNΔsccQ, and ΔsccMΔsccN SCCs showed a markedly lower phosphate content, consistent with our earlier report that the ΔsccN mutant is GroP-deficient[8,9]. Importantly, the phosphate content in the ΔsccM, ΔsccMΔsccQ and ΔsccMΔsccP SCCs was also very low, similar to that of the SCCs from strains deleted for sccN (Fig. 3G), suggesting that SccM also participates in the formation of the GroP acceptor site. Interestingly, the ΔsccQ SCCs contain modestly ( ~ 50%) reduced levels of phosphate (Fig. 3G).

Additionally, we conjugated the isolated SCC variants with a fluorescent dye, 7-amino-1,3-naphthalenedisulfonic acid (ANDS) by reductive amination and examined the electrophoretic mobilities of ANDS-SCCs. The dye ANDS inserts a negative charge to SCCs providing electrophoretic mobility to neutral polysaccharides lacking the GroP moieties as we described in our earlier study[9]. Electrophoresis of ANDS-SCCs isolated from WT and ΔsccP revealed a distinctive "laddering" of SCC bands indicating a highly variable content of negatively charged residues and heterogeneity in charge density. These data are consistent with the presence of GroP modification (Fig. 3H). In agreement with our previously reported analysis of ANDS-SCCs isolated from ΔsccN and ΔsccNΔsccP[9], these GroP-deficient mutants migrated as a single band. The ΔsccM, ΔsccMΔsccQ, ΔsccNΔsccQ, ΔsccMΔsccN and ΔsccMΔsccP SCCs also migrated as a single band, but the ΔsccQ SCCs showed the characteristic "laddering" pattern similar to the WT and ΔsccP SCCs (Fig. 3H). Thus, these data indicate that only the Glc side-chains provided by SccM and SccN serve as the acceptors for GroP. We attribute reduced incorporation of phosphate in the ΔsccQ SCC to the decreased level of the dominant Glc side-chains in this mutant, →2)-α-L-Rhap-(1→3)[α-D-Glcp-(1→2)]-α-L-Rhap-(1→, as we mentioned earlier.

### GroP is linked to position 6 on α-Glc located at position 2 on the 3-linked Rha

A previous analysis of SCC isolated from *S. mutans* after release by autoclaving was conducted by ¹H and ¹³C NMR spectroscopy revealing two polymers, a minor one containing the branched trisaccharide repeating unit →2)-α-L-Rhap-(1→3)[α-D-Glcp-(1→2)]-α-L-Rhap-(1→ and a major one containing the linear rhamnan backbone with the structure →2)-α-L-Rhap-(1→3)-α-L-Rhap-(1→[14]. Significantly, this reported analysis did not detect the GroP modifications or the Glc side-chains linked to the 4-position of branched Rha residues. To investigate the site of attachment of GroP to SCC and characterize the chemical nature of the discovered 4-Rha glucosyl modifications, we purified undegraded SCCs from the cell walls of WT, ΔsccM, ΔsccQ, ΔsccN and ΔsccP, and

two double mutants, ΔsccNΔsccP and ΔsccMΔsccQ, followed by reduction with sodium borohydride, as we described in our earlier study[43], and analyzed them by 1D and 2D NMR spectroscopy. The anomeric region of the ¹H NMR spectrum of the WT SCC was similar to previously reported rhamnan polysaccharides[14]. The 2D ¹H,¹³C-HSQC NMR spectrum revealed, inter alia, a characteristic cross-peak at $\delta_H/\delta_C$ 5.06/98.27 originating from the anomeric atoms of the α-(1→2)-linked glucosyl group as a side-chain. A ³¹P NMR spectrum of the WT SCC showed a resonance at 1.1 ppm, indicative of a phosphodiester linkage[44]. This was also the case for the ΔsccP and ΔsccQ SCCs, whereas ³¹P NMR signals were absent in spectra from the ΔsccM, ΔsccN and ΔsccMΔsccQ SCCs. Additional information that the WT, ΔsccP and ΔsccQ SCCs contained a distinct ³¹P NMR resonance came from 2D ¹H,³¹P-HMBC NMR experiments that showed heteronuclear correlations, presumably over three bonds, between the ³¹P resonance at 1.1 ppm and protons resonating at ~4.23, ~4.08, ~3.96 and ~3.89 ppm (Fig. 4A), consistent with a phosphodiester linkage. In ¹³C NMR spectra, the resonances at $\delta_C$ 71.6 and 67.3 were doublets with $J_{CP}$ of 7.5 and 5.6 Hz, respectively, supporting the findings of a phosphodiester linked substituent. Thus, NMR analysis agrees with phosphate analysis indicating that a phosphodiester linkage requires the Glc side-chain provided by SccN and SccM.

Further analysis of ¹H,¹³C-HSQC and ¹H,¹³C-HMBC NMR spectra of the WT SCC identified a glycerol residue with NMR chemical shifts $\delta_{H1}$ 3.95 and 3.89, $\delta_{C1}$ 67.3, $\delta_{H2}$ 3.92, $\delta_{C2}$ 71.6, $\delta_{H3}$ 3.69 and 3.62, and $\delta_{C3}$ 63.0, similar to those found in GAC[8]. In the ¹H,¹³C-HSQC spectrum, cross-peaks from a methylene group were observed at $\delta_H/\delta_C$ 4.22/64.8 and 4.07/64.8, i.e., at the ¹H NMR chemical shifts identified by correlations in the ¹H,³¹P-HMBC spectrum. From a ¹H,¹³C-HSQC-TOCSY NMR spectrum with a mixing time of 200 ms, a complete spin-system of seven resonances could be identified from $\delta_C$ 64.8, viz., at 5.05, 4.22, 4.07, 3.94, 3.77, 3.64 and 3.57 ppm (Fig. 4B), on the one hand, and from $\delta_{H1}$ 5.06 at 98.3, 73.4, 73.0, 72.1, 70.3 and 61.4 ppm, on the other. Thus, the former spin-system identifies the glycerol residue as being linked to position 6 of the Glc side-chain residue and the latter as the non-substituted Glc side-chain, where the correlations in the latter spin-system are in agreement with the earlier analysis of SCC[14] as well as an NMR chemical shift prediction by CASPER[45] of the trisaccharide repeating unit of the WT SCC. Furthermore, the downfield NMR chemical shift displacement at C6 from 61.4 to 64.8 ppm is consistent with phosphorylation at this position[46]. The trisaccharide repeating unit of the WT SCC was further supported by correlations in the ¹H,¹³C-HMBC NMR spectrum, viz., between $\delta_C$ 98.3 and $\delta_H$ 4.18, $\delta_H$ 5.06 and $\delta_C$ 76.6, 73.4, 73.0, as well as between $\delta_H$ 5.32 and $\delta_C$ 75.4[14]. Small chemical shift displacements were present between residues in the most abundant trisaccharide repeating unit of the WT SCC, lacking GroP, and the minor repeat unit, containing GroP at position 6 of Glc. These can be observed, for example, in the ¹H,¹H-NOESY NMR spectrum with a mixing time of 250 ms where correlations were detected from the anomeric proton of the glucosyl residue at 5.06 ppm to 4.18 and 3.54 ppm (major repeat unit), whereas from 5.05 ppm correlations were observed to 4.16 and 3.57 ppm (minor repeat unit). Furthermore, from the anomeric proton of the branched Rha residue at 5.12 ppm, an intra-residual NOE correlation to H2 at 4.18 ppm (major) was detected, whereas from 5.10 ppm the corresponding correlation was observed to 4.16 ppm (Fig. 4C). Thus, the Glc side-chain residue is partially substituted at position 6 by a GroP residue, which results in a repeating unit having the structure →2)-α-L-Rhap-(1→3)[α-D-Glcp6P(S)Gro-(1→2)]-α-L-Rhap-(1→ (Fig. 4D), the absolute configuration of the substituent is defined by *sn*-Gro-1-P (vide supra), as found in the GAC[8].

### Minor glucosyl substitutions of SCC repeat unit

Our glycosyl linkage analysis of the WT SCC revealed the presence of minor alternative Rha branch points formed by the addition of Glc to the 4-position of Rha. These novel, unexpected modifications could be

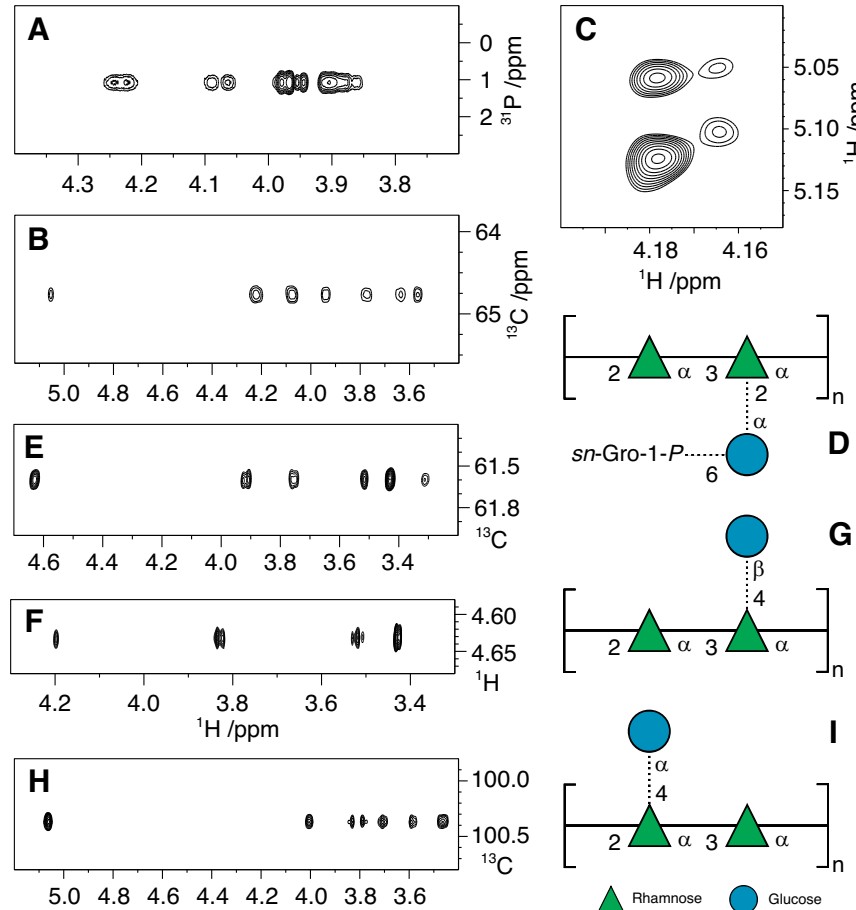

**Fig. 4 | Selected regions of NMR spectra and molecular models of alternative glucosyl-containing trisaccharide repeating units (RU) of SCCs. A** $^1$H,$^{31}$P-HMBC NMR spectrum of the WT SCC identifying phosphodiester-linked entities. **B** Spectral region from a $^1$H,$^{13}$C-HSQC-TOCSY NMR experiment (mixing time of 200 ms) showing the seven-proton glucosyl spin-system of the α-D-Glc$p$6$P$($S$)Gro-(1→2)-linked side-chain residue in the WT SCC. **C** Spectral region from a $^1$H,$^1$H-NOESY NMR experiment at 700 MHz (mixing time of 250 ms) depicting cross-peaks from anomeric protons in the trisaccharide repeating unit (RU) of WT SCC to H2 at ~4.18 ppm of the branched rhamnosyl residue (major) and the corresponding ones at ~4.16 ppm (minor) in which the RU contains an α-D-Glc$p$6$P$($S$)Gro-(1→2)-linked side-chain residue. **D** Canonical structure of the major RU of WT SCC, where the dashed lines denote partial substitution by side-chain entities. **E** $^1$H,$^{13}$C-HSQC-

TOCSY NMR spectrum (τ$_{mix}$ 200 ms) showing the spin-system of the β-D-Glc$p$-(1→4)-linked side-chain residue in the Δ$sccN$ SCC. **F** Spectral region from a $^1$H,$^1$H-NOESY NMR experiment at 700 MHz (τ$_{mix}$ 200 ms) with cross-peaks from anomeric proton of the β-D-Glc$p$-(1→4)-linked side-chain residue in the Δ$sccN$ SCC. The cross-peak at 4.20 ppm originates from H2 of the 2-linked rhamnosyl residue in the backbone of the SCC (cf. the 3D model in Fig. S5). **G** Canonical structure of the RU containing the β-D-Glc$p$-(1→4)-linked side-chain residue, detected in the Δ$sccN$ SCC. **H** Spectral region from a $^1$H,$^{13}$C-HSQC-TOCSY NMR experiment (τ$_{mix}$ 200 ms) showing the seven-proton glucosyl spin-system of the α-D-Glc$p$-(1→4)-linked side-chain residue in the Δ$sccM$ SCC. **I** Canonical structure of the RU containing the α-D-Glc$p$-(1→4)-linked side-chain residue, detected in the Δ$sccM$ SCC. Sugar residues are depicted in SNFG-format with Rha as a green triangle and Glc as a blue circle.

attributed to the expression of specific glycosyltransferases encoded by the SCC genetic locus. To confirm the presence of these branching residues and to investigate the chemical nature of these minor modifications, high resolution 2D $^1$H,$^{13}$C-HSQC NMR analysis was conducted using highly-enriched, borohydride-reduced, SCCs from strains deleted for SccN and SccM, which mediate the synthesis of the dominant →2)-α-L-Rha$p$-(1→3)[α-D-Glc$p$-(1→2)]-α-L-Rha$p$-(1→ repeat unit. NMR analysis of the Δ$sccN$ SCC confirmed that the major form of the polymer is a polysaccharide with a linear repeating →2)-α-L-Rha$p$-(1→3)-α-L-Rha$p$-(1→ backbone. Importantly, NMR spectroscopic analysis of the Δ$sccM$ SCC revealed that a β-linked Glc residue is present (Fig. 4E) as a side-chain substituting the 3-linked Rha backbone residue at position 4, cf. NOE from H1 in **C** to H4 in **A** (Fig. 4F, Table 3, non-primed residues). Interestingly, an additional NOE between H1 in **C** and H2 in **B** of the 2-linked Rha is consistent with a three-dimensional oligosaccharide model (Supplementary Fig. 5) of this structural region of the polysaccharide, further corroborating the proposed trisaccharide structure, →2)-α-L-Rha$p$-(1→3)[β-D-Glc$p$-(1→4)]-α-L-Rha$p$-(1→, as a constituent of the polysaccharide (Fig. 4G). Moreover, the $^{13}$C

NMR chemical shifts for C3 and C4 of the 3,4-disubstituted residue **A** are in excellent agreement with those of a corresponding $O$-methyl rhamnosyl residue, substituted in position 3 by α-L-Rha$p$ and in position 4 by β-D-Glc$p$[47], like in the Δ$sccN$ SCC. In summary, 2D NMR analysis of the Δ$sccN$ and Δ$sccM$ SCC variants confirm and establish the presence of β-Glc at the 4-position of 3-Rha in the polymer. Although it was not possible to directly determine the chemical nature of the 4-Glc substituent of the 2,3,4-substituted Rha (since the 2-substituent is not present in either the Δ$sccN$ or Δ$sccM$ SCCs), it has very likely the β-anomeric configuration.

Because glycosyl linkage analysis of the Δ$sccM$ SCC variant showed a prominent 2,4-Rha branch (Figs. 3C), 2D NMR analysis was employed to characterize this unexpected branched Rha in the Δ$sccM$ and Δ$sccM$Δ$sccQ$ SCCs. In the $^1$H,$^{13}$C-HSQC spectrum of the Δ$sccM$Δ$sccQ$ SCC, the spectral region for anomeric resonances showed cross-peaks, inter alia, at δ$_H$/δ$_C$ 5.20/101.6 and 4.98/102.7 (major) and, inter alia, at δ$_H$/δ$_C$ 5.21/101.2 and 5.06/100.3 (minor) (Supplementary Fig. 6a), similar to those in the Δ$sccM$ SCC (Supplementary Fig. 6b) for which the NMR resonance assignments of the linear rhamnan

**Table 3 | ¹H and ¹³C NMR chemical shifts (ppm) of SCC RPS in D$_2$O at 323 K referenced to TSP ($\delta_H$ 0.00) and dioxane in D$_2$O ($\delta_C$ 67.40); ¹$J_{C1,H1}$ in parenthesis are given in hertz**

| Residue | Label | 1 | ¹$J_{CH}$ | 2 | 3 | 4 | 5 | 6 | 6 |
|---|---|---|---|---|---|---|---|---|---|
| →3,4)-α-L-Rha$p$-(1→2) | A | 4.97 | 172 | 4.19 | 4.04 | 3.83 | 3.84 | 1.34 | 3.92 |
| | | 102.6 | | 70.6 | 80.9 | 78.0 | 68.9 | 17.9 | |
| →2)-α-L-Rha$p$-(1→3) | B | 5.16 | 172 | 4.20 | 3.95 | 3.53 | 3.89 | 1.32 | |
| | | 101.8 | | 79.5 | 70.8 | 73.1 | 70.2 | 17.5 | |
| β-D-Glc$p$-(1→4) | C | 4.63 | 163 | 3.32 | 3.52 | 3.43 | 3.43 | 3.75 | |
| | | 103.7 | | 74.1 | 76.8 | 70.5 | 76.7 | 61.6 | |
| →3)-α-L-Rha$p$-(1→2) | A′ | 4.98 | 173 | 4.16 | 3.86 | 3.58 | 3.78 | 1.29 | |
| | | 102.7 | | 70.7 | 78.4 | 72.4 | 70.1 | 17.4 | |
| →2)-α-L-Rha$p$-(1→3) | B′ | 5.19 | 173 | 4.09 | 3.96 | 3.51 | 3.83 | 1.33 | |
| | | 101.6 | | 78.8 | 70.8 | 73.1 | 70.0 | 17.5 | |
| →3)-α-L-Rha$p$-(1→2) | A″ | 4.98 | 172 | 4.19 | 3.87 | 3.58 | 3.78 | 1.29 | 3.83 |
| | | 102.8 | | 70.7 | 78.4 | 72.4 | 70.1 | 17.4 | |
| →2,4)-α-L-Rha$p$-(1→3) | B″ | 5.21 | 174 | 4.09 | 4.08 | 3.57 | 3.98 | 1.42 | |
| | | 101.2 | | 79.2 | 69.4 | 82.2 | 69.1 | 17.9 | |
| α-D-Glc$p$-(1→4) | C″ | 5.06 | 170 | 3.58 | 3.70 | 3.46 | 4.00 | 3.78 | |
| | | 100.4 | | 72.4 | 73.6 | 70.3 | 72.7 | 61.3 | |

Repeating unit residues of the minor form of Δ$sccN$ are non-primed (A – C). Repeating unit residues of the major form of Δ$sccM$ (rhamnan backbone) are primed (A′ and B′) and those from the minor form are double primed (A″ – C″).

backbone (Table 3, primed residues) were in good agreement with those reported in the literature[14]. Notably, the minor form was shown to originate from a trisaccharide-containing repeating unit (RU) in which the rhamnan backbone carries an α-Glc residue that substitutes the position 4 of the 2-linked Rha in the backbone of the polysaccharide (Fig. 4H, Table 3, double primed residues). The assignments of the NMR resonances from this trisaccharide RU are in excellent agreement with the ¹H and ¹³C NMR chemical shifts predicted by the CASPER program[45] (Supplementary Fig. 7) substantiating the proposed structure (Fig. 4I).

Further analysis of NMR spectra from the WT SCC, aided by spectral assignments obtained from the Δ$sccM$ SCC (Table 3, double primed residues), revealed key resonances such as those traced from H6 (1.42 ppm) in the side-chain substituted backbone residue →2,4)-α-L-Rha$p$-(1→3). In addition, similar connectivities were observed from a peak at 1.44 ppm in the ¹H NMR spectrum, with correlations to 17.8 ppm in the ¹H,¹³C-HSQC spectrum, to 69.1 ppm in the ¹H,¹³C-H2BC spectrum, and to 82.7 ppm in the ¹H,¹³C-HMBC spectrum. The latter ¹³C resonance also showed a correlation to a ¹H resonance at 5.03 ppm in the same spectrum. These results indicate that adjacent to the trisaccharide structural element (Fig. 4I), the neighboring RUs of the rhamnan may also, besides being non-substituted, be substituted by the 2-linked Glc residue and/or substituted by the α-D-Glc$p$6$P$($S$)Gro entity. Among the analyzed mutants, the NMR spectra of the Δ$sccP$ (Supplementary Fig. 6c) and Δ$sccQ$ (Supplementary Fig. 6d) SCCs were similar to those of the WT SCC (Supplementary Fig. 6e). However, in the Δ$sccQ$ SCC, the linear rhamnan backbone polysaccharide was the major form, and the branched trisaccharide RU was the minor form (Supplementary Fig. 6d). Furthermore, ¹H NMR analysis did not detect the presence of the α-linked Glc side-chain components in the Δ$sccN$ SCC (Supplementary Fig. 6f) nor in the Δ$sccN$Δ$sccP$ SCC (Supplementary Fig. 6g). Thus, the NMR data together with the results of a glycosyl linkage analysis point to the function of SccN in providing Glc for the transfer to the 4-position of 2-Rha.

## Major and minor side-chain decorations are important for *S. mutans* morphology

We observed that overnight cultures of the WT and Δ$sccP$ bacteria grown in THY medium remain in suspension. In contrast, the Δ$sccM$,

Δ$sccQ$ and Δ$sccM$Δ$sccQ$ bacteria sediment rapidly (Fig. 5A). Previously, we detected this phenotype in Δ$sccN$ and Δ$sccH$ which we attributed to alterations of the physical properties of the cell wall due to increased intermolecular association between unmodified SCC chains[9]. These data indicate that the minor Glc decorations installed by SccQ also contribute to these intermolecular interactions.

The Glc side-chains provided by SccN and the GroP moieties added by SccH were shown to regulate cell division of *S. mutans*[9]. We compared the morphology of exponentially grown cells of WT, Δ$sccP$, Δ$sccM$ and Δ$sccQ$ using scanning electron microscopy (SEM) (Fig. 5B). The Δ$sccM$ mutant displayed an irregular cell shape and size. In many cells, the division septa were not perpendicular to the long axis of the cell (Fig. 5B). The morphological phenotype of this mutant was similar to the phenotypes reported for Δ$sccH$ and Δ$sccN$[9] which is in agreement with the role of SccM/SccN enzyme pair in installing α-Glc side-chains for GroP decorations. The deletion of *sccP* did not affect the cell shape of *S. mutans*. Surprisingly, the Δ$sccQ$ mutant also showed irregular cell shape, and the cells were significantly smaller and wider than the WT cells (Fig. 5B, C, and Supplementary Table 3). Thus, our data indicate that the major and minor side-chain decorations added by SccM and SccQ are important to support the proper morphology of *S. mutans* cells.

## GroP modification of SCC is critical for biofilm synthesis

*S. mutans* promotes dental caries by adhering to teeth surfaces and participating in production of oral biofilm[2]. When sucrose is present in the growth medium, the bacterium is known to develop an exopolysaccharide-based biofilm[48]. In the absence of sucrose, many *S. mutans* strains are capable to form a protein-based biofilm mediated by protein–protein interactions of the surface proteins[49]. To understand the role of SCC decorations in the pathogenesis of *S. mutans*, we analyzed the formation of the exopolysaccharide- and protein-based biofilms in WT and the mutants defective in Region 2 genes of the SCC gene operon, Δ$sccH$, Δ$sccN$, Δ$sccP$, Δ$sccM$ and Δ$sccQ$. All mutants produced the exopolysaccharide-based biofilms similar to WT when bacteria were grown in the presence of sucrose as demonstrated by crystal violet staining of adhered biomass (Supplementary Fig. 8). However, Δ$sccH$, Δ$sccN$ and Δ$sccM$ were deficient in the protein-based biofilms when bacteria were grown in the presence of glucose (Fig. 5D).

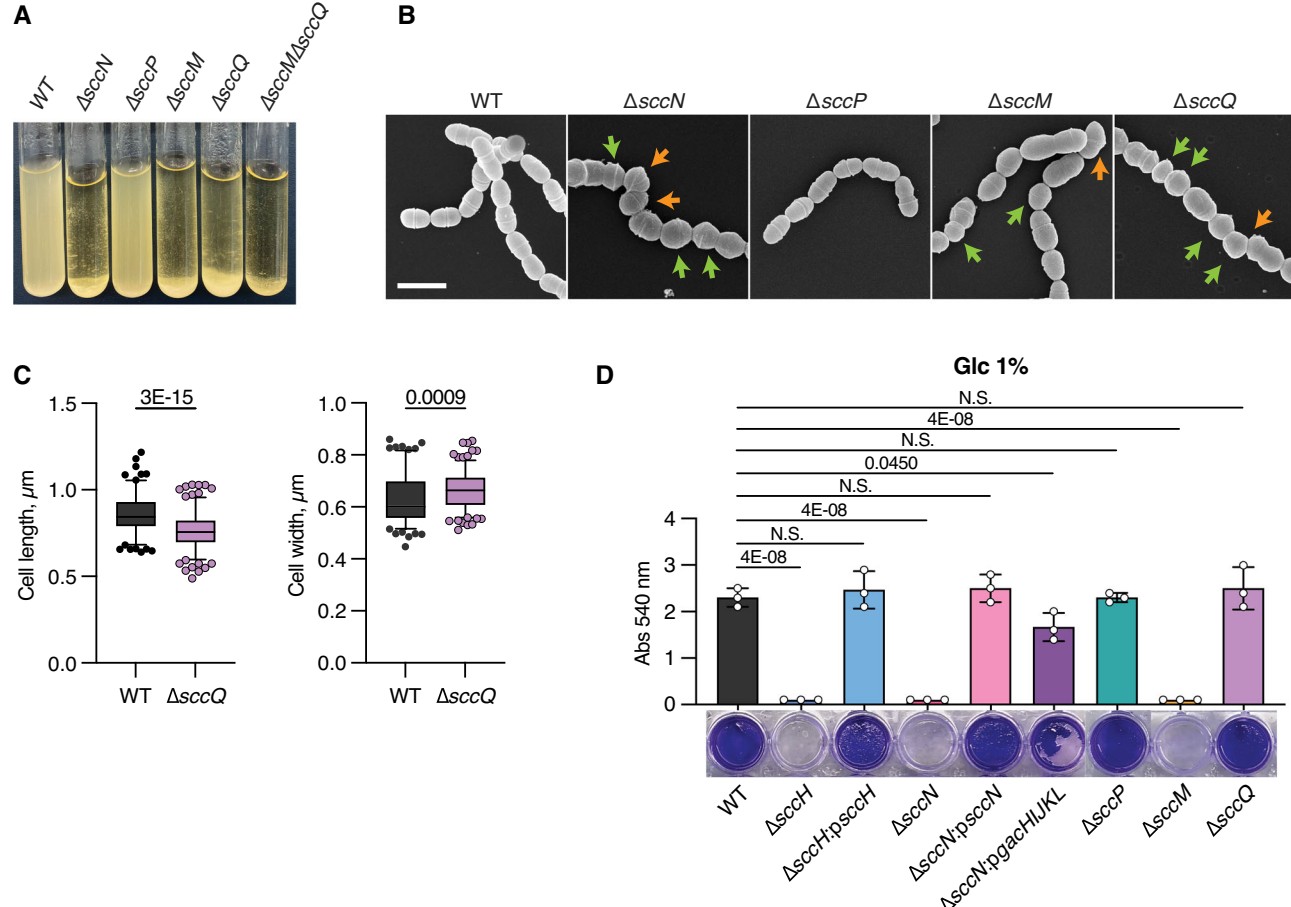

**Fig. 5 | Phenotypes of *S. mutans* strains defective in the SCC side-chain decorations. A** Sedimentation phenotype of *S. mutans* WT and specific mutants after overnight growth in THY broth. **B** Scanning electron micrographs of *S. mutans* WT, Δ*sccN*, Δ*sccP*, Δ*sccM* and Δ*sccQ*. Exponentially growing bacteria were fixed, dehydrated stepwise, and viewed by scanning electron microscopy (SEM). Orange arrows denote the cells with skewed division planes. Green arrows denote cells with irregular size. Scale bar is 1 µm. Representative images from at least three independent experiments are shown in **A** and **B**. **C** Distribution of cell length of cell population at mid-log growth phase for WT and Δ*sccQ*. DIC images of bacterial cells were analyzed using the software ImageJ, ObjectJ plugin to quantify cell length and width. Total number of cells was $n = 144$ for WT, $n = 198$ for Δ*sccQ*. Box plots show the median value (middle line), and 25%, 75% quartiles (boxes), and whiskers represent 5 – 95 percentile. *P* values were determined by unpaired two-tailed t-test with Welch correction. Cell size analysis is presented in Supplementary Table 3. **D** Protein-based biofilm formation by *S. mutans* strains. Biofilms were incubated in UFTYE medium supplemented with 1% D-glucose for 24 h at 37 °C in presence of 5% $CO_2$ and analyzed as outlined in Methods by crystal violet assay (Abs 540 nm). Column and error bars represent the mean ± S.D. of three independent biological experiments. Significant differences ($P < 0.05$) are determined by one-way ANOVA with Dunnett's multiple comparisons test. Image of biofilms formed by *S. mutans* strains after crystal violet staining is representative image of three independent biological experiments. Source data for **D** are provided as a Source data file.

The phenotypes of Δ*sccH* and Δ*sccN* were restored to WT in mutants complemented with the WT copies of *sccH* and *sccN*, respectively. These data indicate that the GroP moieties, and the major Glc side-chains that provide the sites for GroP residues, are essential for biofilm formation.

The side-chains of a homologous polysaccharide GAC are GlcNAc decorated with GroP, and they are encoded by the *gacHIJKL* genes[38]. To replace the major Glc side-chains of SCC with the GAC side-chains, Δ*sccN* was complemented with the *gacHIJKL* genes on a plasmid. The production of the protein-based biofilm by Δ*sccN* was partially restored by this complementation (Fig. 5D) indicating that *S. mutans* biofilm formation is GroP-dependent and independent of the specific glycosyl side-chain.

## Discussion

Our recent analyses of the *S. mutans* and *S. pyogenes* rhamnopolysaccharides, SCC and GAC, established that the side-chain branches on these polymers carry negatively charged GroP moieties[8,9]. In this study, we re-examine the structure of *S. mutans* SCC revealing that the

polysaccharide has major and minor Glc side-chains attached to the polyrhamnose backbone. The major side-chains are α-Glc linked to position 2 of the 3-substituted Rha and partially modified with GroP at the C6 position. Minor side-chains are β-Glc linked to position 4 of the 3- and 2,3-disubstituted Rha and α-Glc linked to position 4 of the 2-substituted Rha. This structural information is critical for a functional description of the enzymes of the SCC biosynthetic pathway and understanding the function of GroP in streptococci.

Our previous work established that decoration of SCC with the GroP moieties requires the SccN glycosyltransferase which is essential for the formation of the Glc side-chain acceptor sites for GroP transfer catalyzed by SccH[8,9]. Here we demonstrate that SccN is a β-Glc-P-Und synthase which works in pair with the SccM glucosyltransferase to transfer α-Glc from β-Glc-P-Und to polyrhamnose. This inverting mechanism, in which a sugar moiety is transferred with inversion of stereochemistry at the anomeric carbon atom of the donor substrate, is a common feature of all GT-C fold enzymes identified to date[50]. Furthermore, we identify SccP as an α-Glc-P-Und synthase which works in pair with the SccQ glucosyltransferase to add β-Glc to the alternative

position 4 on the 3-substituted Rha forming the 3,4-Rha branch (and on 2,3-substituted Rha, to form a 2,3,4-Rha branch). Analysis of the *S. mutans* gene clusters involved in the synthesis of the serotype *f* and *e* rhamnopolysaccharides supports the proposed enzymatic activities of SccN, SccP, and SccQ. The serotype *f* and *e* rhamnopolysaccharides are decorated with α-Glc and β-Glc side-chains, respectively[14,26,29]. An SccN homolog (99% sequence identity) is present in the serotype *f*. In contrast, serotype *e* expresses the homologs of SccP and SccQ (84% and 98% sequence identity, respectively) (Fig. 1A). Surprisingly, the SccN enzyme also provides β-Glc-P-Und for an unknown glycosyltransferase to add α-Glc to position 4 of the 2-substituted Rha forming the 2,4-Rha branch. It is possible, that the side-chains installed by the alternative mechanisms present as minor decorations on SCCs because the dominant α-Glc side-chains attached by SccM may prevent SccQ from adding β-Glc to an alternative site on the same Rha. Furthermore, addition of α-Glc at position 4 of the 2-linked Rha might be repressed due to the competition of SccM and the unknown glycosyltransferase for β-Glc-P-Und. We also noticed that the deletion of SccQ which installs the minor Glc side-chains, reduces the levels of the dominant α-Glc decorations. A clue for why this occurs might be accumulation of the SccP product, α-Glc-P-Und, which sequesters Und-P in a "dead-end" intermediate. In bacteria, Und-P is maintained at low levels in the cytoplasmic membrane (approximately$10^5$ Und-P molecules/cell[51]), and its availability forms a critical bottleneck for the synthesis of essential cell envelope components such as peptidoglycan and cell wall polysaccharides[52,53]. Accordingly, the Und-P in the "dead-end" intermediate cannot be re-cycled and consequently accumulates in the cell, reducing the cellular Und-P levels available for polysaccharide glucosylation as well as peptidoglycan synthesis.

Importantly, when SCCs from various mutant strains are labelled at the reducing end with a fluorescent tag (ANDS) and separated by SDS-PAGE, only strains containing active SccN and SccM show the distinctive 'laddering' pattern attributed to the presence of the GroP modifications. This observation correlates with NMR analysis of SCCs indicating that the major α-Glc side-chains attached to 2,3-Rha serve as the acceptors for GroP. Additionally, we detected a decreased level of phosphate in the ΔsccQ SCCs. Since this polysaccharide contains reduced levels of the major Glc side-chains, this defect is in agreement with the proposed attachment site for the GroP moieties.

As established in our earlier study, defects in GroP decorations lead to severe cell shape alterations and increased autolysis[9]. These phenotypes are attributed to mislocalization of the major autolysin AtlA involved in the separation of daughter cells, and the cell division protein MapZ, which regulates the correct positioning of the Z-ring[9]. Here, we report that the SccM-deficient mutant resembles the *S. mutans* SccH and SccN mutants[9] producing irregularly shaped cells with misplaced division septa. This observation is in line with our conclusion that SccM works in pair with SccN to add α-Glc residues to SCC that are specifically recognized by SccH for GroP transfer. Interestingly, SEM experiments revealed that the SccQ-deficient mutant produces less elongated cells than the parental strain. Given that inactivation of SccQ likely depletes the free Und-P pool as explained above, the underlying mechanisms of morphological abnormalities may be related to inhibition of peptidoglycan synthesis. In both Gram-negative and Gram-positive bacteria, a shortage of Und-P leads to aberrant cell morphologies due to sensitivity of peptidoglycan synthesis to changes in the Und-P pool[52,53]. Thus, our results highlight the importance of the alternative GT-C type glycosyltransferase SccQ in maintaining normal cell wall synthesis and shape. Another important finding of our study is that the GroP decorations are necessary for development of the protein-based *S. mutans* biofilm. Interestingly, this phenotype does not require the minor Glc side-chains and is not specific to a glycosyl modification providing sites for GroP attachment. How GroP regulates *S. mutans* biofilm is still unclear and warrants future studies.

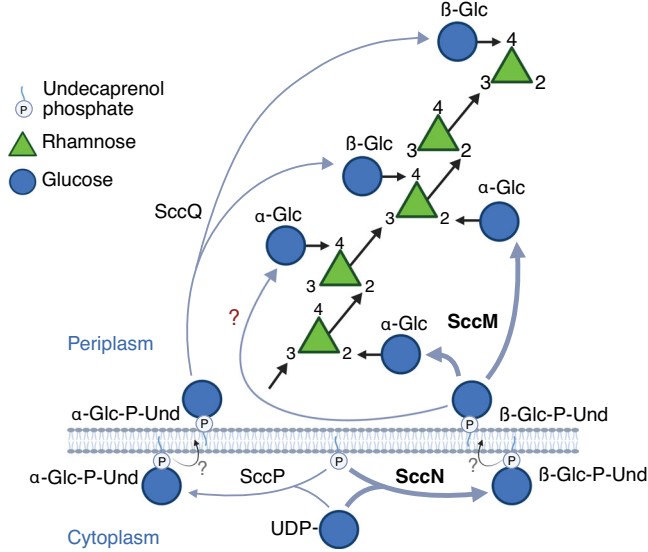

**Fig. 6 | Illustration of the proposed functions of SccN, SccP, SccM, SccQ and a hypothetical glucosyltransferase in modification of the polyrhamnose backbone with Glc side-chains.** In this proposed model, α-Glc-P-Und and β-Glc-P-Und are synthesized on the cytoplasmic surface of the plasma membrane from Und-P and UDP-Glc, catalyzed by GT-A type glycosyltransferases encoded by SccP and SccN, respectively. Following synthesis, the Und-P linked intermediates diffuse transversely to the exoplasmic surface of the plasma membrane, mediated by hypothetical flippase(s)/scramblase(s) (indicated by question marks), where they function as glucosyl donors in glucosylation reactions catalyzed by GT-C type glycosyltransferases, SccQ, SccM and an additional unidentified transferase. SccM uses β-Glc-P-Und to catalyze α-glucosylation of the 2-position of 3-Rha; SccQ uses α-Glc-P-Und to catalyze β-glucosylation of the 4-position of 3-Rha and of 2,3-Rha. α-glucosylation of the 4-position of 2-Rha appears to be catalyzed by a glycosyltransferase encoded outside of the SCC genetic locus, using β-Glc-P-Und, synthesized by SccN, as glucosyl donor. Created with BioRender. Korotkov K. (2024) https://BioRender.com/a74t849.

Taken together, we propose a revised generic structure of SCC containing additional side-chain modifications and a mechanistic model of SCC biosynthesis which identifies the glucosyltransferases involved in the major and alternative pathways of glucosylation (Fig. 6). In this model, polyrhamnose is synthesized on the cytoplasmic face of the plasma membrane and exported to the exoplasmic surface by an ABC transporter complex, as reported for GAC synthesis in *S. pyogenes*[8]. Glc residues are then transferred from β-Glc-P-Und and α-Glc-P-Und synthesized by SccN and SccP on the cytoplasmic surface, respectively, to the 2- and 4-positions of Rha by periplasmically oriented GT-C type transferases, SccM and SccQ (Supplementary Fig. 9), and a prospective third, currently unidentified, glucosyltransferase. Further studies will be required to identify flippase(s)/scramblase(s) involved in the transport of the α-Glc-P-Und and β-Glc-P-Und lipids to the exoplasmic leaflet of the membrane and the glycosyltransferase involved in the transfer of the minor α-Glc side-chains. Because SccN, SccM and SccH are important for biofilm formation, our characterization of SCC biosynthesis may be helpful in guiding the design of anti-caries agents targeting the activities of this pathway in *S. mutans*.

## Methods

### Bacterial strains, growth conditions and media

All plasmids and strains used in this study are listed in Supplementary Table 4. *S. mutans* strains were grown in BD Bacto Todd-Hewitt broth supplemented with 1% yeast extract (THY) or THY agar at 37 °C with 5% $CO_2$ (without aeration). For SCC and phospholipid isolation, *S. mutans*

strains were grown at 37 °C without 5% CO$_2$. To study biofilm formation, *S. mutans* were grown in UFTYE (2.5% tryptone and 1.5% yeast extract) containing either 1% (wt/vol) D-sucrose or 1% (wt/vol) D-glucose at 37 °C in the presence of 5% CO$_2$. *E. coli* strains were grown in Lysogeny Broth (LB) medium or on LB agar plates at 37 °C. When required, antibiotics were included at the following concentrations: chloramphenicol at 10 µg mL$^{-1}$ for *E. coli*; spectinomycin at 500 µg mL$^{-1}$, erythromycin at 5 µg mL$^{-1}$ and kanamycin at 300 µg mL$^{-1}$ for *S. mutans*.

### Construction of mutant strains

All primers are listed in Supplementary Table 5. To delete *sccM* and *sccQ* in *S. mutans* Xc we used a PCR overlapping mutagenesis approach, as previously described[8]. Briefly, 600-700 bp fragments both upstream and downstream of the gene of interest were amplified with designed primers that contained 16-20 bp extensions complementary to the nonpolar antibiotic resistance cassette. The nonpolar spectinomycin, kanamycin, or erythromycin resistance cassettes were PCR-amplified from pLR16T, pOSKAR, and pHY304 (Supplementary Table 4), respectively. The two fragments of the gene of interest and the fragment with the antibiotic resistance cassette were purified using the QIAquick PCR purification kit (Qiagen) and fused by Gibson Assembly (SGA-DNA) using the specific primers. The assembled DNA fragments were directly transformed into the *S. mutans* Xc cells by electroporation. The transformants were selected on THY agar containing the corresponding antibiotic. Double-crossover recombination was confirmed by PCR and Sanger sequencing. The double deletion mutants, Δ*sccM*Δ*sccQ*, Δ*sccM*Δ*sccN* Δ*sccN*Δ*sccQ* and Δ*sccM*Δ*sccP*, were constructed using a similar approach by deleting *sccM* or *sccQ* in the Δ*sccQ*, Δ*sccN* or Δ*sccP* mutant backgrounds. All deletion mutants were verified by whole-genome sequencing (Supplementary Table 6).

### Construction of the plasmids for *E. coli* expression of SccN and SccP

To create vectors for expression of SccN and SccP, the genes were amplified from *S. mutans* Xc chromosomal DNA. The PCR products were digested by the corresponding restriction enzymes and subcloned into a pBAD33 vector. The resultant plasmids, pSccN and pSccP were transferred into competent *E. coli* JW2347 strain that has a deletion of the *gtrB* gene and is therefore devoid of Glc-P-Und synthase activity[54].

### Solubilization and partial purification of recombinant *S. mutans* Glc-P-Und synthases from *E. coli*

For expression of SccN and SccP, *E. coli* JW2347 cells carrying pSccN and pSccP plasmids were grown to an OD$_{600}$ of 0.8 and induced with 13 mM L-arabinose at 25 °C for approximately 3 hours. The cells were lysed in 20 mM Tris-HCl pH 7.5, 300 mM NaCl with two passes through a microfluidizer cell disrupter. The lysate was centrifuged (1000 g, 15 minutes, 4 °C) followed by centrifugation of the supernatant (40,000 g, 60 min, 4 °C) to isolate the membrane fraction. The SccN and SccP enzymes were solubilized from the cell pellets and partially purified. Solubilization mixtures contained 50 mM HEPES, pH 7.4, 1× bacterial protease inhibitor cocktail (Pierce Chemical Co.), 10 mM DTT, 0.5% CHAPS (3-[(3-cholamidopropyl)dimethyl-ammonio]-1-propane sulfonate) and *E. coli* membrane fraction (5 mg/ml protein). After 1 h on ice, solubilization mixtures were centrifuged (100,000 g, 30 min) and the supernatant liquid removed and loaded onto a 30 mL column of DEAE 650 M (Toso Haas), equilibrated in the cold in 10 mM HEPES, pH 8.0, 20% glycerol, 5 mM DTT, 0.1% CHAPS. The column was eluted with 2 column volumes of buffer and then with a 40 mL linear gradient of NaCl (0 - 1 M). Fractions of 3 mL were collected and analyzed for protein and Glc-P-Und synthase activity. Fractions containing Glc-P-Und synthase activity were combined, concentrated by ultrafiltration (Amicon, 10,000 MWCO), snap-frozen on dry-ice ethanol in aliquots and stored at −20 °C until use.

### Preparation of membrane fractions from *S. mutans* cells

For preparation of membrane fractions, *S. mutans* cells were grown to a final OD$_{600}$ of 0.6-0.8 in 400 mL cultures of THY-broth at 37 °C and collected by centrifugation at 8,000 g, 20 min. The cells were resuspended in 0.1 M HEPES-OH, pH 8.0, 0.25 M sucrose and 10 mM MgCl$_2$ and digested with mutanolysin (0.1 U/mL) at 37 °C, 1 hr. Following sensitization with mutanolysin, the digests were diluted with 40 mL ice-cold H$_2$O and lysed by beat-beating with a Braun homogenizer in the cold (5 min). Following lysis, the homogenate was decanted from the beads, and supplemented with 0.01 mL DNase I (10 mg/mL) and sucrose to a final concentration of 0.25 M. After incubation on ice for 30 min, homogenates were centrifuged (1,000 g, 10 min), and the pellet was discarded. The supernatant was sedimented (40,000 g, 30 min), and the pellet was resuspended in 10 mM HEPES-OH, 0.25 M sucrose, re-sedimented (40,000 g, 30 min), resuspended in 1 mL buffer (10 mM HEPES-OH, 0.25 M sucrose) and stored in 0.1 mL aliquots at -20 °C until analysis.

### Preparation of undecaprenyl phosphate (Und-P)

Und-P was synthesized from undecaprenol (ARC, ARCD 0126) by the trichloroacetimidate procedure as described in[55] and purified by preparative TLC on glass-backed Analtech Silica Gel HL sheets.

### In vitro analysis of Glc-P-Und synthase activity

Reactions for the in vitro analysis of Glc-P-Und synthase activity contained 50 mM HEPES-OH pH 7.4, 20 mM MgCl$_2$, 10 mM DTT, 0.5 mM sodium orthovanadate, the indicated amount of UDP-[$^3$H]Glc (50-500 cpm/pmol) and bacterial enzyme (either *S. mutans* membrane suspension or solubilized *E. coli* membrane proteins) in a total volume of 0.02 mL. Where indicated, reactions were supplemented with the indicated amount of Und-P, dispersed in 1% CHAPS by bath sonication, and with a final concentration of 0.35% CHAPS. Following incubation at 37 °C for 3-30 min, the reaction was stopped by the addition of 2 mL CHCl$_3$/CH$_3$OH (2:1). Reactions were freed of unincorporated radioactivity as described previously[38] and analyzed for radioactivity by scintillation spectrometry or by TLC on 5 x 20 cm glass-backed Analtech Silica Gel HL sheets developed in CHCl$_3$/CH$_3$OH/H$_2$O/NH$_4$OH (65:30:4:1). Reactions with membrane fractions from *S. mutans* were subjected to mild alkaline de-acylation with 0.1 N KOH in methanol/toluene (3:1) for 60 min at 0 °C, to destroy glycerolipids and freed of released radioactivity by partitioning as described[38]. Degradation of Glc-lipids by mild acid and mild alkaline treatment were conducted as previously described[38].

### Discharge of synthetic [$^3$H]Glc-P-Und by reversal of Glc-P-Und synthase

Large amounts of the [$^3$H]Glc-P-Und$_{SccN}$ and [$^3$H]Glc-P-Und$_{SccP}$ were synthesized using Und-P, UDP-[$^3$H]Glc and the appropriate partially purified Glc-P-Und synthase. The synthetic [$^3$H]Glc-P-Undecaprenols were purified by Folch partition[56] and preparative TLC on glass-backed Analtech Silica Gel HL sheets developed in CHCl$_3$/CH$_3$OH/H$_2$O/NH$_4$OH (65:30:4:1) as described below. The glycolipids were dispersed by sonication in 1% CHAPS, for 5 min in a bath sonicator, and used as substrates in the discharge reactions. Reaction mixtures were identical to reactions for the in vitro analysis of Glc-P-Und synthase activity, except that Und-P and UDP-Glc were replaced by the indicated purified [$^3$H]Glc-P-Und (10,000 cpm) and UDP (5 mM) and contained the appropriate partially purified Glc-P-Und synthase (SccN Glc-P-Und synthase activity, -0.7 µg protein; -555 pmol/min; SccP Glc-P-Und synthase activity, -0.13 µg protein, 430 pmol/min). Following incubation at 37 °C for the indicated times, the reactions were stopped by the addition of 2 mL CHCl$_3$/CH$_3$OH (2:1) and partitioned with 0.4 mL 0.15 M NaCl, to separate the starting [$^3$H]Glc-P-Undecaprenols from water-soluble product. The organic (lower) phase and aqueous (upper) were separated, transferred to scintillation vials, dried under air and

analyzed for radioactivity by scintillation spectrometry in 4 mL of Econosafe Counting Cocktail (Research Products International), containing 0.2 mL, 1 % SDS.

## Purification of *S. mutans* phosphoglycolipids for ESI-MS analysis

To purify glycolipids for ESI-MS analysis, *S. mutans* cells (2 L) from late exponential phase cultures ($OD_{600}$ ~ 0.8) were recovered by sedimentation (10,000 $g$, 30 min) and washed twice with ice-cold phosphate-buffered saline (PBS). Cells were resuspended in 15 ml PBS, sensitized by incubation with mutanolysin (0.2 M Na acetate, pH 5.2, 200 U/ml, 1 h, 37 °C) and extracted with mixtures of $CHCl_3$/$CH_3OH$ essentially by the method of Bligh-Dyer[57] and freed of water-soluble contaminants by three rounds of aqueous partitioning (Folch)[56]. The organic extract was dried on a vacuum rotary evaporator, dissolved in a small volume of $CHCl_3$/$CH_3OH$ (2:1), transferred quantitatively to a 12.5 × 100 mm screw cap glass tube (with Teflon lined cap) and partitioned with 1/5th volume of water to remove residual salts. The desalted organic extract was dried under a stream of nitrogen gas (at ~ 30 °C) and the glycerolipids were destroyed by deacylation in 4 mL 0.1 M KOH in toluene/$CH_3OH$ (1:3) at 0 °C, 60 min. Following deacylation, the reactions were neutralized with acetic acid, diluted with $CHCl_3$ and 0.9% NaCl/10 mM EDTA to give a final composition of $CHCl_3$/$CH_3OH$/0.9% NaCl (3:2:1). The two-phase mixture was mixed vigorously and centrifuged to separate the phases. The aqueous phase was discarded and the organic phase was washed with ~1/3 volume of $CHCl_3$/$CH_3OH$/0.9% saline (3:48:47), two times. The organic phase was dried under a stream of $N_2$ gas, dissolved in 1 mL $CHCl_3$/$CH_3OH$ (2:1), partitioned with 1/5th volume of water and dried again. The dried glycolipid fraction was redissolved in 1 mL of $CHCl_3$/$CH_3OH$ (19:1) and applied to a column of silica gel (BioRad) equilibrated in $CHCl_3$. The column was washed with 5 column volumes of $CHCl_3$ and 5 column volumes of acetone. The glycophospholipids were eluted with 5 column volumes of $CHCl_3$/$CH_3OH$ (2:1), partitioned with 1/5th volume of water and dried under a stream of $N_2$ gas. Fractions recovered from silica gel chromatography were analyzed by TLC on silica gel, developed in $CHCl_3$/$CH_3OH$/$H_2O$/$NH_4OH$ (65:25:4:1) and visualized by exposure to iodine vapors. Comparison of mobility on silica with that of authentic [$^3$H]Glc-P-Und, synthesized separately, confirmed that the glycolipids were quantitatively recovered in the $CHCl_3$/$CH_3OH$ (2:1) fraction following silica gel chromatography. The glycolipid fraction was dried and purified further by preparative thin layer chromatography on silica gel G developed in $CHCl_3$/$CH_3OH$/$H_2O$/$NH_4OH$ (65:25:4:1). Glc-P-Unds were visualized by exposure to iodine vapors and located by reference to [$^3$H]Glc-P-Und standard in adjacent lanes detected by a Bioscan 2000 radiochromatoscanner. Glc-P-Unds were recovered from the silica gel plate by elution with $CHCl_3$/$CH_3OH$ (2:1), dried under a stream of nitrogen gas and reserved for analysis by LC/MS/MS.

## Mass spectrometry analysis of phospholipids isolated from *S. mutans* strains

The glycolipids were analyzed by LC-MS using a Q-exactive mass spectrometer, employing polarity switching and data dependent acquisition, and an Ultimate 3000 ultra high-performance liquid chromatography system (Thermo Fisher Scientific, San Jose, CA) on a Kinetex C18 reversed-phase column (2.1 mm × 100 mm, 2.6 μm, Phenomenex, USA). As described previously[38], a two-component gradient elution was performed employing solvent A, acetonitrile/water (2:3, v/v) containing 10 mM ammonium formate and 0.1 % formic acid and solvent B, isopropyl alcohol/acetonitrile (9:1, v/v) containing 10 mM ammonium formate and 0.1 % formic acid. Column temperature was maintained at 40 °C, and the flow rate was set to 0.25 mL/min. Mass spectrometric detection was performed by electrospray ionization in negative ionization mode with source voltage maintained at 4.0 kV. The capillary temperature, sheath gas flow and auxiliary gas

flow were set at 330 °C, 35 and 12 arbitrary units, respectively. Full-scan MS spectra (mass range m/z = 400 to 1500) were acquired with a resolution R = 70,000 and ABC target 5e5. MS/MS fragmentation was performed using high-energy C-trap dissociation with resolution R = 35,000 and AGC target 1e6. The normalized collision energy was set 30.

## Preparation of *S. mutans* cell walls

*S. mutans* cell wall was isolated from late exponential phase cultures ($OD_{600}$ ~ 0.8) by the SDS-boiling procedure as described for *S. pneumoniae* and *S. pyogenes*[38,58]. Purified cell wall samples were lyophilized and stored at −20 °C before the analysis.

## Isolation of SCCs from *S. mutans* cell walls

SCCs were released from highly enriched cell wall preparations by mild acid hydrolysis, following chemical *N*-acetylation, as described previously[9], and partially purified by size exclusion chromatography on a column of Bio Gel P150 (Bio Rad, 1 × 18 cm) equilibrated in 0.2 N Na acetate, pH 3.7, 0.15 M NaCl[43].

## Analysis of SCCs for composition and glycosyl linkages

Composition of sodium borohydride-reduced SCCs was determined as trimethylsilyl (TMS) derivatives of O-methyl glycosides following methanolysis in methanol/1 N HCl (100 °C, 3 h) either, in house as described previously[43], or at the Complex Carbohydrate Research Center, University of Georgia (CCRC). Quantities of component sugars were calculated from gas-chromatographic (GC) areas of component TMS-methylglycosides after normalization to the internal standard and response factor corrections.

Response factors (Rf) were calculated from standard mixtures of sugars and internal standard using the formula: $Rf = (A_x/A_{is})/(C_x/C_{is})$ in which, $A_x$ is GC area of analyte, $A_{is}$ is GC area of internal standard, $C_x$ is amount of analyte and $C_{is}$ is amount of internal standard. Amounts of sugars were calculated using the formula: $C_x = C_{is} \times (A_x/A_{is})/Rf$. For illustrative purposes, the amounts of sugars are expressed as nmol of each sugar in the sample normalized to 50 nmol Rha (the proposed size of the polyrhamnose backbone). In some instances, an exponential increase in response factor associated with increasing amounts of analyte relative to internal standard was observed. To correct for this change in response factor, standard curves of increasing amounts of analytes to a constant quantity of internal standard were prepared and used to estimate the appropriate response factor, when necessary.

Linkage analysis of the glycans was determined at the CCRC as partially methylated alditol acetates as described[59]. Quantitative estimates of specific linkages per polymer were calculated by multiplying the nmol sugar per 50 nmol Rha in each sample, by the relative area percent of the GC area for each linkage type.

## Phosphate assays

Total phosphate content in SCCs was determined by the malachite green method[60] following digestion with perchloric acid. Fractions containing 10 to 80 μL were heated to 110 °C with 20 μL 70% perchloric acid (Fisher Scientific) in 13 × 100 mm borosilicate screw-cap disposable culture tubes for 1 h. The reactions were diluted to 160 μL with water and 100 μL was transferred to a flat-bottom 96-well culture plate. Malachite Green reagent (0.2 mL) was added and the absorbance at 620 nm was read after 10 min at room temperature. Malachite Green reagent contained one vol 4.2% ammonium molybdate tetrahydrate (by weight) in 4 M HCl, 3 vol 0.045% malachite green (by weight) in water and 0.01% Tween 20. Phosphate concentrations were determined using a phosphate standard curve.

## Modified anthrone assay

Total Rha and Glc contents were estimated using a minor modification of the anthrone procedure, as described elsewhere[9]. Briefly, reactions

contained 0.08 mL of aqueous sample plus water and 0.32 mL. anthrone reagent (0.2 % anthrone in concentrated $H_2SO_4$). The samples were heated to 100 °C, 10 min, cooled in water bath (room temperature), 0.3 mL was transferred into a plastic, clear-bottom 96-well plate and the absorbance at 580 nm and 700 nm was recorded. The chromophore produced by Rha has a relatively discreet absorbance maximum at 580 nm, but is essentially zero at 700 nm. The contribution of Glc to absorbance at 580 nm can be estimated from the absorbance at 700 nm, employing pure Glc standard solutions, and subtracted from the absorbance of the unknown samples at 580 nm to obtain the Rha-specific signal. The Rha and Glc concentrations were estimated using L-Rha and D-Glc standard curves, respectively.

### PAGE analysis of SCCs derivatized with ANDS

SCCs were derivatized with 7-amino-1,3-naphthalenedisulfonic acid (ANDS) by reductive amination of the reducing end of the polysaccharide as described previously[9]. Derivatized SCCs were further purified by SEC over a Bio Gel P150 (Bio Rad, 1 × 18 cm) equilibrated in 0.2 N Na acetate, pH 3.7, 0.15 M NaCl[43]. ANDS-labeled SCCs were resolved on Bolt™ Bis-Tris 4-12% Gel (NW04125, ThermoFisher) in Bolt™ MOPS SDS running buffer at 200 V for 32 min, and visualized by fluorescence.

### NMR spectroscopy

SCCs (5-27 mg, dry weight) isolated from WT and the various mutant strains were dissolved in $D_2O$ (0.55 mL). NMR experiments were conducted in 5 mm outer diameter NMR tubes on Bruker NMR spectrometers operating at $^1$H frequencies of 400 or 700 MHz at temperatures of 23 °C or 50 °C, respectively, using experiments suitable for resonance assignments of glycans[44,61]. $^1$H NMR chemical shifts were referenced to internal sodium 3-trimethylsilyl-(2,2,3,3-$^2$H$_4$)-propanoate ($\delta_H$ 0.0), $^{13}$C chemical shifts were referenced to external dioxane in $D_2O$ ($\delta_C$ 67.4) and $^{31}$P chemical shifts were referenced to external 2% $H_3PO_4$ in $D_2O$ ($\delta_P$ 0.0). Acquired NMR data were processed and analyzed using the TopSpin® software from Bruker.

### Scanning electron microscopy (SEM)

Exponentially growing bacteria ($OD_{600}$ ~ 0.7) were fixed with paraformaldehyde (4% final concentration), and then pipetted onto microscope slide cover glasses coated with poly-L-lysine. Following one hour incubation at room temperature, the cover glasses were washed three times with PBS. Bacteria were dehydrated stepwise in a gradient series of ethanol (35%, 50%, 70%, 80% and 96% for 20 min each at room temperature and then 100% overnight at −20 °C), followed by critical point drying with liquid $CO_2$ in a Leica EM CPD300. Samples were coated with about 5 nm of platinum controlled by a film-thickness monitor. SEM images were performed in the immersion mode of an FEI Helios Nanolab 660 dual beam system.

### Differential interference contrast (DIC) microscopy

Exponentially growing bacteria ($OD_{600}$ ~ 0.7) were fixed with paraformaldehyde (4% final concentration), pipetted onto microscope slide cover glasses (high performance, D = 0.17 mm, Zeiss) coated with poly-L-lysine, and allowed to settle for one hour at room temperature. The samples were washed three times with PBS and mounted on a microscope slide with ProLong Glass Antifade (Invitrogen). Samples were imaged on a Leica SP8 equipped with 100 × , 1.44 N.A. objective, DIC optics. ImageJ software ObjectJ plugin was used to measure the sizes of cells.

### Biofilm assay

UFTYE supplemented with 1% (wt/vol) sucrose or glucose was inoculated with the 17 h overnight cultures of *S. mutans* strains creating a 1:100 fold inoculation. Bacterial biofilms were grown in 24-well plates (Corning, NY) at 37 °C in the presence of 5% $CO_2$. After 24 h of growth, the

cell suspension in each well was removed and biofilm was washed three times by immersing the microtiter plate in deionized water. The biofilm was stained with crystal violet (0.2% w/v in water) followed by three washes and de-staining in 10% methanol, 7.5% acetic acid solution. Finally, the absorbance of the de-staining solution was measured at 540 nm.

### Statistical analysis

Unless otherwise indicated, statistical analysis was carried out on pooled data from at least three independent biological repeats. Statistical analyses were performed using Graph Pad Prism version 9.2.0. Quantitative data was analyzed using one-way ANOVA, 2-way ANOVA, and unpaired t-test as described for individual experiments. A *P*-value equal to or less than 0.05 was considered statistically significant.

### Reporting summary

Further information on research design is available in the Nature Portfolio Reporting Summary linked to this article.

## Data availability

All data generated during this study are included in the article and Supplementary Information files. Source data are provided with this paper.

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

## Acknowledgements

The authors thank Dr. Catherine Chaton (University of Kentucky) for the help with figure preparation and Dr. Christopher D. Radka (University of Kentucky) for helpful discussion of ESI-MS analysis of phosphoglycolipids. This work was supported by NIH grants R01 DE028916 from the NIDCR and R01 AI143690 from the NIAID (to N.K.), and the Swedish Research Council (no. 2022-03014) and The Knut and Alice Wallenberg Foundation (to G.W.). Scanning electron microscopy was performed at the Electron Microscopy Center, which belongs to the National Science Foundation NNCI Kentucky Multiscale Manufacturing and Nano Integration Node, supported by ECCS–1542174. Carbohydrate composition/linkage analysis at the Complex Carbohydrate Research Center was supported by the Chemical Sciences, Geosciences and Biosciences Division, Office of Basic Energy Sciences, U.S. Department of Energy grant DE-SC0015662 (to P.A.), as well as by NIH grant R24 GM137782 from the NIGMS (to P.A.). The content is solely the responsibility of the authors and does not necessarily represent the official views of the National Institutes of Health.

## Author contributions

J.S.R., G.W., C.H., P.A., and N.K. designed the experiments. J.S.R., S.Z., N.R.M., C.W.K. and I.B. performed functional and biochemical experiments. S.Z. performed microscopy analysis. P.D. performed MS analysis. G.W. performed NMR studies. S.Z., K.V.K. and N.K. constructed plasmids and isolated mutants. J.S.R., S.Z., P.D., I.B., C.H., A.J.M., K.V.K., G.W. and N.K. analyzed the data. N.K., J.S.R. and G.W. wrote the manuscript with contributions from all authors. All authors reviewed the results and approved the final version of the manuscript.

## Competing interests

The authors declare no competing interests.
