## [Transparent Peer Review file · Nature Communications]

Structure and mechanism of biosynthesis of *Streptococcus mutans* cell wall polysaccharide

Corresponding Author: Dr Natalia Korotkova

Version 0:

Reviewer comments:

Reviewer #1

(Remarks to the Author)

The manuscript "Structure and mechanism of biosynthesis of *Streptococcus mutans* cell wall polysaccharide" by Jeffrey S. Rush et al. provides an important contribution to the structural characterization and biological implications of the polyrhamnose-glucose polymer of the cariogenic bacterium *S. mutans*. The study is based on the construction of defined glycosyltransferase mutants from the corresponding gene cluster to elucidate the roles of selected enzymes in the biosynthesis pathway, focusing on the so far not-well defined glucose and GroP modifications. While the NMR part of the study is sound and well explained, there are several points the authors may wish to consider to improve the quality of the manuscript.

The study is of interest for glycobiologists, microbiologists, and oral biologist, and pinpoints potential novel target points for the design of anti-cariogenic strategies.

General points:

- The manuscript is written in a rather inhomogeneous style with regards to the accuracy in the description of methods. The degree of details given for the different methods varies a lot. Some methods are only referenced, while for others, unnecessary details are given. Please unify the style of the Methods section and add appropriate references. In the current state, it would not be always possible to repeat the experiments.
- Restructuring the text in terms of moving information from the Results to the Introduction section would help to improve the readability of the manuscript
- In several places references are missing or insufficient.
- The Discussion section is repetitive in several instances and should be streamlined.
- Replace "rhamnopolysaccharide" by rhamnose-glucose polymer (RGP)
- Proteins should start with capital letters and not be written in italics, e.g., p. 5, L. 9.
- Replace "dramatic" by "clear" throughout the manuscript.
- Replace "delta xxx strain by "delta xxx mutant" throughout the manuscript, e.g., p. 7., l. 23, 25, 26.

Abstract:

- Change sentence to: SCC consists of a polyrhamnose backbone of $\rightarrow 3)\alpha\text{-Rha}(1\rightarrow 2)\alpha\text{-Rha}(1\rightarrow$ repeats with glucose (Glc) side-chains and glycerol phosphate (GroP) decorations
- L. 12: Is b-Glc correct? If there is a major and a minor b-Glc modification, this has to be explained before.
- The reference style does not conform with the style recommended by Nat. Commun.

Introduction:

- Page 2: l. 17: add reference
- L. 18: Dental plaque is never formed by *S. mutans* alone; it is a polymicrobial community. Rephrase and add references.
- Change l. 18-25 to: Similar to other streptococci, *S. mutans* strains decorate peptidoglycan with rhamnose (Rha) and glucose (Glc)-containing cell wall polysaccharides, so called rhamnose-glucose polymers (RGP) that are important for viability and biofilm formation of these bacteria 1. The RGPs are functional homologs of teichoic acids which are linked via a phosphodiester bond to the cell wall peptidoglycan of many Gram-positive bacteria 2, 3, 4. Streptococcal RGPs have Cite: Kovacs, C. J., Faustoferrri, R. C., Bischer, A. P., & Quivey, R. G., Jr. (2019). *Streptococcus mutans* requires mature rhamnose-glucose polysaccharides for proper pathophysiology, morphogenesis and cellular division. *Molecular Microbiology*, 112(3), 944–959. <https://doi.org/10.1111/>

mmi.14330

Kovacs, C. J., Faustoferri, R. C., & Quivey, R. G., Jr. (2017). RgpF is required for maintenance of stress tolerance and virulence in *Streptococcus mutans*. *Journal of Bacteriology*, 199(24), e00497–17. <https://doi.org/10.1128/jb.00497-17>

- Page 3: Replace l. 3-4. By: *S. mutans* strains are classified into four serotypes, c, e, f and k, based on variations in the structures of their RGP's 19. These have distinct functions in cell division and morphology.

Cite: Reviewed in doi:10.3389/fcimb.2024.1357631

- L. 19: It would be of interest to know which procedure during preparation might have affected the loss of GroP. Add a reference, describing the preparation of GAC

- P. 4, L. 12: Add reference

- L. 25: Add reference. Or was this shown in the present study?

Results:

- p.4/5: "In bacteria, multi-component transmembrane glycosylation complexes catalyze the attachment of sugars to a variety of glycans. This machinery requires a GT-A fold glycosyltransferase to synthesize an Und-P linked glycosyl donor, a flippase to move the lipid intermediate across the cell membrane and a periplasmically-oriented GT-C fold glycosyltransferase to transfer the sugar from the lipid intermediate to the nascent polysaccharide 29, 30." This is not a result. It should be included in the Introduction section.

- L. 15: It is unclear, if for this experiment, the recombinant enzymes or the recombinant strain was used. Information of the recombinant enzyme production (e.g., yield, set-up of assays) is missing.

- L. 17: Define "under these reaction conditions"

- L. 24: Please provide analytical evidence for the statement: "consistent with Glc-P-Und"

- P. 6 L. 14-15: "Although the enzymatic products formed by SccN and SccP from UDP-Glc and Und-P were chromatographically indistinguishable". Please show these results

- L. 17-18: "Glycosyltransferases producing products with a sugar 1-phosphate (like Glc-P-Und) are known to be readily reversible". Rephrase sentence for clarity

- L. 21: "we tested the purified products of each reaction". How was the purity of the products ascertained? Which analysis was done? Please provide data.

- P. 7, L. 7-12: "During fragmentation of sugar 1-phosphates by ESI-MS, if the phosphodiester at C1 and the hydroxyl group at C2 are oriented in trans to the sugar ring then [Und-HPO₄]⁻ (m/z =845.7) is released as the highest molecular weight fragment (Fig. 2d)". Rephrase sentence for clarity

- P. 10, l. 15: Replace "somewhat" by scientific terminology

- L. 18-20: Move to Methods section

- L. 21-22: "distinctive "laddering" of SCC bands indicating a high content of negatively charged residues and heterogeneity in charge density". I assume that the ladder-like pattern is due to the differences in the polymerization degree of the Rha-repeats in the backbone. Rephrase for clarity

- L. 25: "GroP-deficient mutants migrated as a single band." This would implicate that GroP modification affect the degree of polymerization? Please comment on this.

- P. 11, l. 2: "SCC isolated from *S. mutans* after release by autoclaving...". How does autoclaving release a pure SCC that can be subjected to NMR analysis? Please add a corresponding reference

- P. 13, l. 11: Should be: "NMR spectroscopic analysis"

- P. 14, l. 26: Refer to figure

- P. 15, l. 7: Remove "Planktonic"

- L. 19: Add reference

- P. 16, L. 15.-16.: "All mutants produced the exopolysaccharide-based biofilms similar to WT (Supplementary Fig. 8). However, *deltascch*, *deltascchN* and *deltascchM* were deficient in the protein-based biofilms (Fig. 5d)." State briefly in the text, how this was determined.

Discussion

- L. 21-23: "it was assumed that the cell walls of most streptococci do not carry the net negative charge attributed to the phosphate-rich polymers" Why was this assumed? Add references. In Gram-positive bacteria, there are several other charged cell wall polymers in addition to teichoic acid. Please account for this in the Discussion section.

- L. 22: Add reference

- P. 17: L. 17, 18: Specify "identity"

- P. 18, l. 25-26: "Glc residues are then transferred from beta-Glc-P-Und and alpha-Glc-P-Und.." These intermediates need to be first flipped to the periplasm. This step is missing in the pathway explanation.

Methods

- P. 19, l. 4: Move primer table to the next section where mutants are constructed

- L. 9-11. Why were different cultivation conditions used for liquid culture and agar plates? *S. mutans* grows differently under aerobic and CO₂-conditions.

- L. 11: Specify biofilm conditions more precisely.

Please add strain designation, as different strains of *S. mutans* behave naturally differently with regard to biofilm formation.

- P. 20, L. 2. "using a similar approach" Describe how this was done or cite a reference.

- L. 11: "deletion of the *gtrB* gene 42" Why was this *E. coli* deletion mutant chosen as an expression host?

- P. 20. L. 25: "SCCs isolated from *S. mutans* wilt-type (WT) and the mutant strains were dissolved in D₂O." State concentration of SCCs and total amount of material used for NMR analysis.

- P. 21, l. 5. "malachite green method" Add reference

- L. 14. "Phosphate concentrations were determined"

- L. 23-24: "Derivatized SCCs were further purified by SEC over a Superdex 75 10/300 GL column (GE Healthcare Bio-Sciences AB). " Add details of the SEC purification
- L. 24-25: „ANDS-labeled SCCs were resolved on Bolt™ Bis-Tris 4-12% Gel (ThermoFisher) and visualized by fluorescence." Add information about PAGE conditions and gel dimensions/apparatus.
- L. 29-30: "TLC on silica gel G". Add description glass, aluminium?) and provider of TLC plates
- P. 22: Section "Solubilization and partial purification of *S. mutans* Glc-P-Und synthases" Move this section up right below the description of overexpression
- L. 11. "and *E. coli* membrane proteins (5 mg/ml)." It is unclear which *E. coli* proteins are meant here.
- L. 24: Where is this indicated?
- L. 15: "*S. mutans* membrane suspension" The preparation of Smu membranes has not been described.
- L. 31: Specify scintillation spectrometry
- P. 26, l. 4: Add references for SEM and sample preparation
- L. 24: Add reference for CV assay

Reviewer #2

(Remarks to the Author)

The Manuscript by Rush et al. details the role of SccN, SccP, SccM and SccQ in the decoration of SCC with glucose and glycerol phosphate. In doing so the manuscript describes new structure for SCC with one major and two minor Glc modifications. The manuscript uses a range of methods to characterise the roles of the four proteins, including recombinant protein expression and genetic knockouts, coupled with carbohydrate structural analysis by MS and NMR.

Minor comments

Fig 2.A No units on the Y-axis

Line 145-147 " Δ SccN membranes was dramatically reduced" Authors should quantify the reduction using the data i.e. 2-fold reduced etc

Major comment

Authors use linkage analysis of the SCC purified from various genetic knockouts to determine the roles of SccN, SccM and SccP and SccQ. Assumptions are made based on the single knockouts that SccN and SccM are the major enzyme pair. Why is there no analysis of the SCC from the double knockout to confirm this?

Reviewer #3

(Remarks to the Author)

This manuscript builds on previous work by the same group (Zamakhaeva et al (2021)) and describes two novel glycosyltransferases found in the biosynthesis gene cluster for serotype c-specific carbohydrates (SCC) of *Streptococcus mutans*. They claim that these glycosyltransferases attach glucose moieties at different positions of the rhamnopolysaccharide, working in tandem with two previously described glucosyl phosphoryl undecaprenol (Glc-P-Und) synthases, which synthesize Glc-P-Und in α - or β -confirmation, respectively. Additionally, the authors claim that the more dominant α -glucosylation attached by the glycosyltransferase sccM is involved in biofilm formation, due to attachment of negatively charged GroP. Altogether, the authors characterize the SCCs of different mutants extensively via NMR and other analytical techniques. The manuscript seems to be overall valid but could profit from additional controls, text clarifications, and simpler description. The following points should be addressed:

1. Line 35: The authors frequently talk about "major" and "minor" Glc modifications. They however do not explain what is meant by that. If it is just describing the amount of Glc on the SCC, that should be stated.
2. Line 234: The authors frequently mention 2,3-substituted Rha, however if I understand correctly, they mean that the Rha is substitute with Glc, but also with the next Rha in the backbone polymer, which makes it difficult to distinguish. – Could this be stated more clearly, to distinguish between backbone linkage and Glc linkage?
3. Fig 3: Especially in the WT, but even in the absence of the dominant sccM glycosyltransferase (can it be called dominant?), β -Glc attachment mediated by sccQ at the C4-position is rather low, as shown in Fig. 3. Why is it so inactive. Could it have other functions than the one suggested in the manuscript?
4. Upon deletion of sccQ, an increase in 2,3-Rha would be expected (Fig 3, D) according to the model proposed in Fig 6. But the opposite is observed. How can this be explained?
5. According to Figure 3E, less than half of all rhamnose units in the polymer are modified with Glc. Could something other than Glc be attached, like the D-ala modification of wall teichoic acid (WTA)?
6. Fig3E: How can it be explained that deletion of sccQ reduces the amount of Glc in SCCs, but deletion of sccP has no effect? Is the model in Figure 6 incomplete or can sccQ use β -Glc-P-Und as additional substrate?
7. The authors claim that no GroP is attached to the sccQ linked β -Glc modification of Rha. Deletion of sccQ however reduces the amount of phosphate in the SCCs (Fig. 3F). The observation that GroP is not transferred to sccQ linked β -Glc is based on SDS-Page of ANDS labeled SCCs which observes that deletion of sccM prevents the addition of charged GroP. However, is it possible that GroP is transferred from 2-linked α -Glc to 4-linked β -Glc? Or could the charge change due to low addition of β -Glc be too small to be visible in the SDS-Page? A Gro measurement as performed in the previous paper by Zamakhaeva et al (2021) should be used.
8. Phages of *S. mutans* are proposed to bind the Glc modifications of SCCs. This could be tested with the new mutants.

Minor:

1. An overview of these serotype specific rhamnopolysaccharides would help the reader. - It could be added to figure 1.
2. Line 184: Is this the m/z for Glc-P-Und? - Could the name be added directly to the mass?
3. Line 202-204: This deduction seems overall logical. - However, could NMR of purified Glc-P-Und confirm these findings?
4. Line 216-217: Are we talking about the linkage unit GlcNAc? Could this be more specific? Could this be added to Fig 6?
5. Line 217: What is the difference between the reducing end GlcNAc and non-reducing end GlcNAc?
6. Line 224: What could be the explanation of this observation? - You would usually expect that Glc contents are not reduced more than half if only one transferase is deleted. Often these glycosyltransferases can even complement for each other.
7. Line 231: Why is there GlcNAcitol found in the SCCs?
8. Line 500-501: Is serotype e only glycosylated with β -Glc? - Or is that unknown?
9. Line 1031: Is it not sccM which attaches the "major form" of glycosylations?
10. Fig 3: nmol Pi per 50 nmol Rha or Pi/50 Rha.
11. Fig 6: Should the SCC not be drawn linked to the membrane via the GlcNAc linker? I assume it is attached at this state.

Version 1:

Reviewer comments:

Reviewer #2

(Remarks to the Author)

Authors have dealt with all comments appropriately.

Reviewer #3

(Remarks to the Author)

In this manuscript, the authors extensively characterize the SCCs of *Streptococcus mutans* via NMR and other analytical techniques. They focus on glycosyltransferases attaching glucose in different conformations at different positions of the polyramnose polymer. Additionally, they characterize two Glc-P-Und synthases which are important in this process and show evidence that the SCCs and the newly characterized modifications are involved in biofilm formation.

The respective scientific field will clearly benefit from the results presented in this manuscript, since several novel aspects about *S. mutans* SCCs are presented in a conclusive and logical manner.

The methodology seems to be overall sound, even though I cannot evaluate the extensive chemical analysis due to a lack of insights into chemical structure analysis of carbohydrates.

The concerns raised during the revision process were sufficiently addressed.

made.

REVIEWER COMMENTS

Reviewer #1 (Remarks to the Author):

The manuscript “Structure and mechanism of biosynthesis of Streptococcus mutans cell wall polysaccharide” by Jeffrey S. Rush et al. provides an important contribution to the structural characterization and biological implications of the polyrhamnose-glucose polymer of the cariogenic bacterium *S. mutans*. The study is based on the construction of defined glycosyltransferase mutants from the corresponding gene cluster to elucidate the roles of selected enzymes in the biosynthesis pathway, focusing on the so far not-well defined glucose and GroP modifications. While the NMR part of the study is sound and well explained, there are several points the authors may wish to consider to improve the quality of the manuscript.

The study is of interest for glycobiologists, microbiologists, and oral biologist, and pinpoints potential novel target points for the design of anti-cariogenic strategies.

General points:

- The manuscript is written in a rather inhomogeneous style with regards to the accuracy in the description of methods. The degree of details given for the different methods varies a lot. Some methods are only referenced, while for others, unnecessary details are given. Please unify the style of the Methods section and add appropriate references. In the current state, it would not be always possible to repeat the experiments.

The authors thank reviewer # 1 for the positive comments and the thoughtful and thorough review of our manuscript. We have made a conscientious effort to address the concerns of the reviewer and feel that this review has improved the manuscript substantially.

The authors are committed to providing sufficient detail in the Methods section to enable the replication of the described experiments, and we apologize if we have failed in this instance. When appropriate, our previously described methods are simply referenced, but in some instances these methods had not been previously described, and we felt it was essential to provide a full description. Assuming that the primary concern here is to provide more detail to some of the methods we have made the following additions.

To the ‘**Modified Anthrone Procedure**’ paragraph we have added:

Reactions contained 0.08 mL of aqueous sample and water and 0.32 mL anthrone reagent (0.2 % anthrone in concentrated H₂SO₄). The samples were heated to 100°C, 10 min, cooled in water bath (room temperature), 0.3 mL was transferred into a 96-well plate and the absorbance at 580 nm and 700 nm was recorded. The chromophore produced by Rha has a relatively discreet absorbance maximum at 580 nm, but is essentially zero at 700 nm. The contribution of Glc to absorbance at 580 nm, in the assay reactions, can be

estimated from the absorbance at 700 nm (using spill-over absorbance values obtained from pure Glc standard solutions) and subtracted from the absorbance at 580 nm to obtain the Rha-specific signal. The Rha and Glc concentrations were estimated using L-Rha and D-Glc standard curves, respectively.

To 'Discharge of synthetic [³H]Glc-P-Und by reversal of Glc-P-Und synthase':

Following incubation at 37 °C for the indicated times, the reactions were stopped by the addition of 2 mL CHCl₃/CH₃OH (2:1) and partitioned with 0.4 mL 0.15 M NaCl, to separate the starting [³H]Glc-P-Undecaprenols from water-soluble product. The organic (lower) phase and aqueous (upper) were separated, transferred to scintillation vials, dried under air and analyzed for radioactivity by scintillation spectrometry in 4 mL of Econosafe Counting Cocktail (Research Products International), containing 0.2 mL, 1 % SDS.

To 'Purification of *S. mutans* phosphoglycolipids for ESI-MS analysis':

This section was modified by removing the detailed description of the lipid extraction and replacing it with references to the methods of Bligh-Dyer and Folch.

- Restructuring the text in terms of moving information from the Results to the Introduction section would help to improve the readability of the manuscript

We thank the reviewer for a careful reading of the manuscript and the desire to improve its readability, but we respectfully disagree that moving background information from the Results section to the Introduction would be of any benefit. The background statement about GT-A and GT-C type transferases is required to explain how we happened to focus on SccN, SccP, SccM and SccQ and to explain the hypothetical basis of their functions.

- In several places references are missing or insufficient.

In the revision, we have provided additional references. However, we refrain from citing the information that we consider common knowledge in the field, when there is no strong rationale for doing so or from the speculative studies/reviews.

- The Discussion section is repetitive in several instances and should be streamlined.

We have streamlined our discussion, largely removing the first sentences.

- Replace "rhamnopolysaccharide" by rhamnose-glucose polymer (RGP)

The term rhamnopolysaccharide has been used in the literature for rhamnose-containing cell wall polysaccharides in streptococci, lactococci and enterococci

(PMID: 36113580, 25035517, 32345640, 31292230, 35105886). We would like to continue using it in our publications because it accurately describes more than one polysaccharide. In addition, the term rhamnose-glucose polymer (RGP) is not correct for *S. mutans* serotype k rhamnose-containing cell wall polysaccharide because it has an α -galactose side-chains instead of Glc. However, we have added a phrase indicating the alternative term. "In *S. mutans* serotype c, the polysaccharide, called the serotype c-specific carbohydrate (SCC) (also referred to elsewhere as rhamnose-glucose polysaccharide or RGP²⁸), is reported to contain α -glucose (Glc) side-chains attached to the 2-position of 3-linked Rha units¹⁴."

- Proteins should start with capital letters and not be written in italics, e.g., p. 5, L. 9.

Corrected (thank you).

- Replace "dramatic" by "clear" throughout the manuscript.

Corrected (thank you).

- Replace "delta xxx strain" by "delta xxx mutant" throughout the manuscript, e.g., p. 7., l. 23, 25, 26.

Corrected (thank you).

Abstract:

- Change sentence to: SCC consists of a polyrhamnose backbone of $\rightarrow 3)\alpha$ -Rha(1 \rightarrow 2) α -Rha(1 \rightarrow repeats with glucose (Glc) side-chains and glycerol phosphate (GroP) decorations

We have changed this sentence to read:

"SCC consists of a polyrhamnose backbone of $\rightarrow 3)\alpha$ -Rha(1 \rightarrow 2) α -Rha(1 \rightarrow repeating disaccharide units, with glucose (Glc) side-chains and glycerol phosphate (GroP) decorations".

- L. 12: Is b-Glc correct? If there is a major and a minor b-Glc modification, this has to be explained before.

This is correct and the discovery of these minor modifications (there are actually two) is central to this manuscript.

- The reference style does not conform with the style recommended by Nat. Commun.

We have used the style recommended by Nat. Commun

Introduction:

- Page 2: l. 17: add reference

The reference is provided (thank you).

- L. 18: Dental plaque is never formed by *S. mutans* alone; it is a polymicrobial community. Rephrase and add references.

We have changed this sentence to read:

“The Gram-positive bacterium *Streptococcus mutans* is a normal inhabitant of the human oral cavity recognized as a major etiological agent of human dental caries¹. Ability of this organism to colonize tooth surfaces forming biofilms is directly associated with the development of dental caries².”

- Change I. 18-25 to: .Similar to other streptococci, *S. mutans* strains decorate peptidoglycan with rhamnose (Rha) and glucose (Glc)-containing cell wall polysaccharides, so called rhamnose-glucose polymers (RGP) that are important for viability and biofilm formation of these bacteria¹. The RGPs are functional homologs of teichoic acids which are linked via a phosphodiester bond to the cell wall peptidoglycan of many Gram-positive bacteria^{2, 3, 4}. Streptococcal RGPs have
Cite: Kovacs, C. J., Faustoferri, R. C., Bischer, A. P., & Quivey, R. G., Jr. (2019). *Streptococcus mutans* requires mature rhamnose-glucose polysaccharides for proper pathophysiology, morphogenesis and cellular division. *Molecular Microbiology*, 112(3), 944–959. <https://doi.org/10.1111/mmi.14330>

Kovacs, C. J., Faustoferri, R. C., & Quivey, R. G., Jr. (2017). RgpF is required for maintenance of stress tolerance and virulence in *Streptococcus mutans*. *Journal of Bacteriology*, 199(24), e00497–17. <https://doi.org/10.1128/jb.00497-17>

Respectfully, we modified this sentence to read:

“Similar to other streptococcal species, *S. mutans* strains decorate peptidoglycan with rhamnose (Rha)-containing cell wall polysaccharides, so called rhamnopolysaccharides, that are important for the viability of these bacteria^{3, 4, 5, 6}. Streptococcal rhamnopolysaccharides are functional homologs of teichoic acid glycopolymers that are present in the cell walls of many Gram-positive bacteria^{7, 8, 9}. Loss of cell wall polysaccharides in *S. mutans* is associated with a pleiotropic phenotype that includes abnormal cell morphology, enhanced cellular autolysis and defective biofilm formation^{5, 10, 11}.”

- Page 3: Replace I. 3-4. By: *S. mutans* strains are classified into four serotypes, c, e, f and k, based on variations in the structures of their RGPs¹⁹. These have distinct functions in cell division and morphology.

Cite: Reviewed in doi:10.3389/fcimb.2024.1357631

Respectfully, we disagree with the citation of the review because this statement in the review is not linked to the relevant study. To our knowledge, there have

been no studies analyzing how variations in the structures of serotype-specific rhamnopolysaccharides affect cell division and morphology of streptococci.

- L. 19: It would be of interest to know which procedure during preparation might have affected the loss of GroP. Add a reference, describing the preparation of GAC

This is referenced in the preceding sentence.

- P. 4, L. 12: Add reference

This statement is a follow-up to the preceding sentence - the reference is in the preceding sentence.

- L. 25: Add reference. Or was this shown in the present study?

We have clarified that this result is shown in the present study:
“Furthermore, we report that the GroP decoration is critically important for *S. mutans* biofilm formation. Our NMR studies reveal that GroP is attached exclusively on the 6-OH of the α -Glc side-chains donated by SccN and SccM”.

Results:

- p.4/5: “In bacteria, multi-component transmembrane glycosylation complexes catalyze the attachment of sugars to a variety of glycans. This machinery requires a GT-A fold glycosyltransferase to synthesize an Und-P linked glycosyl donor, a flippase to move the lipid intermediate across the cell membrane and a periplasmically-oriented GT-C fold glycosyltransferase to transfer the sugar from the lipid intermediate to the nascent polysaccharide 29, 30.” This is not a result. It should be included in the Introduction section.

Respectfully, we would point out that identifying SccN and SccP as GT-A type transferases and SccM and SccQ as GT-C type transferases, and defining their roles in the modification of the polyrhamnose chains, is very much a ‘result’ since this was not known prior to these experiments. We believe that including this statement here is appropriate to the development of this narrative, and it would be out of place, and possibly misinterpreted, in the introduction.

- L. 15: It is unclear, if for this experiment, the recombinant enzymes or the recombinant strain was used. Information of the recombinant enzyme production (e.g., yield, set-up of assays) is missing.

This sentence has been revised to read:
“In membrane fractions prepared from the recombinant strains, both synthases actively catalyzed the formation of a [³H]glucolipid when incubated in vitro with UDP-[³H]Glc and Und-P.”

- L. 17: Define „under these reaction conditions”

'Under these reaction conditions' refers to the specific reaction conditions as defined in the Methods section. We have modified this sentence to read: "However, the apparent maximal rates for SccN are higher than that of SccP (Fig. 2a and Table 1)."

- L. 24: Please provide analytical evidence for the statement: "consistent with Glc-P-Und"

We apologize if this statement is unclear. The sentence has been edited to read: "Preliminary experiments revealed that *S. mutans* membrane fractions synthesize two classes of glucolipids: a major glucolipid product with chromatographic properties (thin layer silica gel G, see Methods) similar to [³H]Glc-P-Und formed in the in vitro reactions with *E. coli* expressed SccN and SccP, described above, and an additional minor glucolipid which is most likely a glucosyldiglyceride."

In text (lines 135-142) we show that SccN and SccP expressed in *E. coli* catalyzed the formation of Glc-P-Und. In experiments to confirm that these two enzymes also catalyze the synthesis of Glc-P-Und in *S. mutans*, we discovered that membrane fractions from *S. mutans* catalyze the formation of two glucolipids (line 148). One that has properties similar to Glc-P-Und and another that appears to be a glucosyldiglyceride. That analytical evidence is presented a few sentences later (lines 157-166): "The activities were stimulated by the exogenous addition of Und-P as a suspension in CHAPS detergent (Supplementary Fig. 1a), and inhibited by amphomycin which is known to form an insoluble complex with Und-P (Supplementary Fig. 1b, c)³¹. Further analysis showed that the [³H]glucolipids are anionic, sensitive to mild acid and resistant to mild alkali as expected for a glycosyl phosphoryl isoprenoid. In addition, a co-migrating compound, purified from the *S. mutans* membrane fraction by organic solvent extraction and preparative TLC, yielded a molecular ion, m/z =1007.7 by ESI-MS which is expected for Glc-P-Und (Supplementary Fig. 1d). Thus, our data indicate that both SccN and SccP catalyze the synthesis of Glc-P-Und."

- P. 6 L. 14-15: "Although the enzymatic products formed by SccN and SccP from UDP-Glc and Und-P were chromatographically indistinguishable". Please show these results

We have included a figure in supplemental material showing these data. (Supplemental Fig 1b).

- L. 17-18: "Glycosyltransferases producing products with a sugar 1-phosphate (like Glc-P-Und) are known to be readily reversible". Rephrase sentence for clarity

This sentence has been reworded: "GT-A type glycosyltransferases are known to be readily reversible in the presence of the appropriate nucleoside diphosphate, re-forming the nucleosidediphospho-sugar and releasing Und-P."

- L. 21: “we tested the purified products of each reaction”. How was the purity of the products ascertained? Which analysis was done? Please provide data.

The substrates ($[^3\text{H}]\text{Glc-P-Und}$ products formed by the individual synthases) of the reverse reactions were radiochemically pure based on TLC analysis. The sentence is reworded:

“To investigate if the enzymatic products of SccN and SccP are interchangeably equivalent substrates in the reverse direction to reform UDP-Glc, we purified the $[^3\text{H}]\text{Glc-P-Und}$ products by preparative TLC and tested them for reverse synthesis by incubation with UDP in the presence of each of the enzymes.”

- P. 7, L. 7-12: “During fragmentation of sugar 1-phosphates by ESI-MS, if the phosphodiester at C1 and the hydroxyl group at C2 are oriented in trans to the sugar ring then $[\text{Und-HPO}_4]^-$ ($m/z = 845.7$) is released as the highest molecular weight fragment (Fig. 2d). “ Rephrase sentence for clarity

The sentence was reworded:

“During fragmentation of sugar 1-phosphates by ESI-MS, if the phosphodiester at C1 and the hydroxyl group at C2 are oriented trans to the sugar ring (i.e., the functional groups reside on opposite sides of the plane of the sugar ring) then $[\text{Und-HPO}_4]^-$ ($m/z = 845.7$) is released as the highest molecular weight fragment (Fig. 2d). However, if the relative orientation of the groups at C1 and C2 is arranged cis (i.e., the functional groups reside on the same side of the plane of the sugar ring), then an additional cross-ring fragmentation product, $[\text{Und-PO}_4\text{-C}_2\text{H}_3\text{O}]^-$ ($m/z = 887.7$) is released (Fig. 2d).”

- P. 10, l. 15: Replace “somewhat” by scientific terminology

The sentence is reworded:

“Interestingly, the ΔsccQ SCCs contain modestly (~50%) reduced levels of phosphate (Fig. 3f).”

- L. 18-20: Move to Methods section

Respectfully, we believe this statement belongs here to explain the origin of the electrophoretic mobility.

- L. 21-22: “distinctive “laddering” of SCC bands indicating a high content of negatively charged residues and heterogeneity in charge density”. I assume that the ladder-like pattern is due to the differences in the polymerization degree of the Rha-repeats in the backbone. Rephrase for clarity

Our conclusion is that this laddering is due primarily to a variable number of charged residues per polymer not to variations in molecular size. The sentence is reworded:

“Electrophoresis of ANDS-SCCs isolated from WT and $\Delta sccP$ revealed a distinctive “laddering” of SCC bands indicating a highly variable content of negatively charged residues and heterogeneity in charge density.”

- L. 25: “GroP-deficient mutants migrated as a single band.” This would implicate that GroP modification affect the degree of polymerization? Please comment on this.

The GroP-deficient mutants migrate as a single band because they have only a single negative charge provided by the ANDS tag (see above).

- P. 11, l. 2: „SCC isolated from *S. mutans* after release by autoclaving...”. How does autoclaving release a pure SCC that can be subjected to NMR analysis? Please add a corresponding reference

The autoclave extraction method is a traditional (although crude and inefficient) method for isolation of microbial polysaccharides (PMID: 32140, 13522509, 21561675, 28971282). The mechanism of polysaccharide release by autoclaving is not understood. Probably the polysaccharide - peptidoglycan linkage unit is unstable to heating/autoclaving.

The following sentence has the relevant citation. “A previous analysis of SCC isolated from *S. mutans* after release by autoclaving was conducted by ^1H and ^{13}C NMR spectroscopy revealing two polymers, a minor one containing the branched trisaccharide repeating unit $\rightarrow 2)\text{-}\alpha\text{-L-Rhap-(1}\rightarrow 3)[\alpha\text{-D-Glcp-(1}\rightarrow 2)]\text{-}\alpha\text{-L-Rhap-(1}\rightarrow$ and a major one containing the linear rhamnan backbone with the structure $\rightarrow 2)\text{-}\alpha\text{-L-Rhap-(1}\rightarrow 3)\text{-}\alpha\text{-L-Rhap-(1}\rightarrow$ ¹⁴”

- P. 13, l. 11: Should be: “NMR spectroscopic analysis”

Corrected (thank you)

- P. 14, l. 26: Refer to figure

New Supplemental Figure 6a, b, c, d, e, f and g has been generated to show a selected region for anomeric resonances in ^1H , ^{13}C -HSQC NMR spectra of WT, $\Delta sccN$, $\Delta sccP$, $\Delta sccM$, $\Delta sccQ$, $\Delta sccN\Delta sccP$ and $\Delta sccM\Delta sccQ$ SCC variants. References to this figure have been provided.

- P. 15, l. 7: Remove “Planktonic”

This sentence has been reworded:

“We observed that overnight cultures of the WT and $\Delta sccP$ bacteria grown in THY medium remain in suspension.”

- L. 19: Add reference

This sentence has been reworded and has relevant citations:

“*S. mutans* promotes dental caries by adhering to teeth surfaces and participating in production of oral biofilm. When sucrose is present in the growth medium, the bacterium is known to develop an exopolysaccharide-based biofilm⁴⁸. In the absence of sucrose, the bacterium is capable to form a protein-based biofilm mediated by protein–protein interactions of the surface proteins⁴⁹.”

- P. 16, L. 15.-16.: “All mutants produced the exopolysaccharide-based biofilms similar to WT (Supplementary Fig. 8). However, *deltascchH*, *deltascchN* and *deltascchM* were deficient in the protein-based biofilms (Fig. 5d).” State briefly in the text, how this was determined.

This sentence has been reworded:

“All mutants produced the exopolysaccharide-based biofilms similar to WT when bacteria were grown in the presence of sucrose as demonstrated by crystal violet staining of adhered biomass (Supplementary Fig. 8). However, $\Delta scchH$, $\Delta scchN$ and $\Delta scchM$ were deficient in the protein-based biofilms when bacteria were grown in the presence of glucose (Fig. 5d).”

Discussion

- L. 21-23: “it was assumed that the cell walls of most streptococci do not carry the net negative charge attributed to the phosphate-rich polymers” Why was this assumed? Add references. In Gram-positive bacteria, there are several other charged cell wall polymers in addition to teichoic acid. Please account for this in the Discussion section.

This sentence has been removed from the Discussion section.

- L. 22: Add reference

Please see our response above.

- P. 17: L. 17, 18: Specify “identity”

These sentences have been modified:

“An *ScchN* homolog (99% sequence identity) is present in the serotype f. In contrast, serotype e expresses the homologs of *ScchP* and *ScchQ* (84% and 98% sequence identity, respectively).”

- P. 18, l. 25-26: “Glc residues are then transferred from beta-Glc-P-Und and alpha-Glc-P-Und..” These intermediates need to be first flipped to the periplasm. This step is missing in the pathway explanation.

In Fig.6, the prospective flippases are indicated by the curved arrows and labelled with ‘?’ to emphasize that they have not yet been identified. In addition, we mentioned an unknown flippase in the proposed mechanism of SCC modification with Glc residues. Please see the following sentence:

“In this model, polyrhamnose is synthesized on the cytoplasmic face of the plasma membrane and exported to the exoplasmic surface by an ABC transporter complex, as reported for GAC synthesis in *S. pyogenes*⁸. Glc residues are then transferred from β -Glc-P-Und and α -Glc-P-Und synthesized by SccN and SccP on the cytoplasmic surface, respectively, to the 2- and 4-positions of Rha by periplasmically oriented GT-C type transferases, SccM and SccQ (Supplementary Fig. 9), and a prospective third, currently unidentified, glucosyltransferase. Further studies will be required to identify flippase(s)/scramblase(s) involved in the transport of the α -Glc-P-Und and β -Glc-P-Und lipids to the exoplasmic leaflet of the membrane and the glucosyltransferase involved in the transfer of the minor α -Glc side-chains.”

Methods

- P. 19, l. 4: Move primer table to the next section where mutants are constructed

The supplementary table 5 (primers) has been moved to the next section.

- L. 9-11. Why were different cultivation conditions used for liquid culture and agar plates? *S. mutans* grows differently under aerobic and CO₂-conditions.

S. mutans Xc does not grow differently in liquid culture with and without CO₂. However, CO₂ stimulates bacterial growth on agar plates. Methods were modified to indicate that CO₂-free conditions were only used to grow bacteria for SCC and phospholipid isolation. We used CO₂-free conditions for these experiments because large volume of bacteria was required.

“*S. mutans* strains were grown in BD Bacto Todd-Hewitt broth supplemented with 1% yeast extract (THY) or THY agar at 37 °C with 5% CO₂ (without aeration). For SCC and phospholipid isolation, *S. mutans* strains were grown at 37 °C without 5% CO₂.”

- L. 11: Specify biofilm conditions more precisely.
Please add strain designation, as different strains of *S. mutans* behave naturally differently with regard to biofilm formation.

All strains used in our studies are listed in the Supplementary Table 4.
Biofilm assay has sufficient details to reproduce the experiment.

- P. 20, l. 2. “using a similar approach “ Describe how this was done or cite a reference.

The description of construction of mutant strains has sufficient details to reproduce the experiment. Double mutants were constructed using a similar approach used for single mutants by deleting respective genes in the mutant backgrounds.

- L. 11: “deletion of the *gtrB* gene 42” Why was this *E. coli* deletion mutant chosen as an expression host?

This sentence has been amended to read:

“The resultant plasmids, pScCN and pScCP were transferred into competent *E. coli* JW2347 strain that has a deletion of the *gtrB* gene and is therefore devoid of Glc-P-Und synthase activity ⁴².”

- P. 20. L. 25: “SCCs isolated from *S.* mutants wilt-type (WT) and the mutant strains were dissolved in D₂O. “ State concentration of SCCs and total amount of material used for NMR analysis.

The concentrations of SCCs and amount of D₂O used for NMR analysis have been provided:

“SCCs (5-27 mg, dry weight) isolated from WT and the various mutant strains were dissolved in D₂O (0.55 mL).”

- P. 21, l. 5. “malachite green method “ Add reference

Reference of malachite green method has been provided.

- L. 14. “Phosphate concentrations were determined”

Corrected (thank you).

- L. 23-24: “Derivatized SCCs were further purified by SEC over a Superdex 75 10/300 GL column (GE Healthcare Bio-Sciences AB). “ Add details of the SEC purification

The description of SEC is corrected to read:

“Derivatized SCCs were further purified by SEC over a Bio Gel P150 (Bio Rad, 1×18 cm) equilibrated in 0.2 N Na acetate, pH 3.7, 0.15 M NaCl ⁴³.”

- L. 24-25: „ANDS-labeled SCCs were resolved on Bolt™ Bis-Tris 4-12% Gel (ThermoFisher) and visualized by fluorescence.” Add information about PAGE conditions and del dimensions/apparatus.

This sentence has been amended to read:

“ANDS-labeled SCCs were resolved on Bolt™ Bis-Tris 4-12% Gel (NW04125, ThermoFisher) in Bolt™ MOPS SDS running buffer at 200 v for 32 min, and visualized by fluorescence.”

- L. 29-30: “TLC on silica gel G”. Add description glass, aluminium?) and provider of TLC plates

The description of silica gel G glass-backed plates has been provided.

- P. 22: Section “Solubilization and partial purification of *S.* mutans Glc-P-Und synthases” Move this section up right below the description of overexpression

The sections in Methods have been moved to mirror the experiments described in Results

- L. 11. "and *E. coli* membrane proteins (5 mg/ml)." It is unclear which *E. coli* proteins are meant here.

Edited to reflect that we refer to both *E. coli* expressed enzymes:

"The SccN and SccP enzymes were solubilized from the cell pellets and partially purified. Solubilization mixtures contained 50 mM HEPES, pH 7.4, 1× bacterial protease inhibitor cocktail (Pierce Chemical Co.), 10 mM DTT, 0.5% CHAPS (3-[(3-cholamidopropyl)dimethyl-ammonio]-1-propane sulfonate) and *E. coli* membrane fraction (5 mg/ml protein)."

- L. 24: Where is this indicated?

This refers to individual experiments, during the kinetic analysis, the concentration of UDP-Glc was varied, in other experiments a standard amount of UDP-Glc was employed. The amounts are in either the figure legend or indicated in the figure itself.

- L. 15: "*S. mutans* membrane suspension" The preparation of Smu membranes has not been described.

We apologize for this omission and have added a paragraph on preparation of *S. mutans* membrane fractions. Please see the section "**Preparation of membrane fractions from *S. mutans* cells**".

- L. 31: Specify scintillation spectrometry

We have added this to the end of the section:

"Following incubation at 37 °C for the indicated times, the reactions were stopped by the addition of 2 mL CHCl₃/CH₃OH (2:1) and partitioned with 0.4 mL 0.15 M NaCl, to separate the starting [³H]Glc-P-Undecaprenols from water-soluble product. The organic (lower) phase and aqueous (upper) were separated, transferred to scintillation vials, dried under air and analyzed for radioactivity by scintillation spectrometry in 4 mL of Econosafe Counting Cocktail (Research Products International), containing 0.2 mL, 1 % SDS."

- P. 26, l. 4: Add references for SEM and sample preparation

The description of SEM and sample preparation is sufficient to reproduce the experiment.

- L. 24: Add reference for CV assay

The description of CV assay is sufficient to reproduce the experiment.

Reviewer #2 (Remarks to the Author):

The Manuscript by Rush et al. details the role of SccN, SccP, SccM and SccQ in the decoration of SCC with glucose and glycerol phosphate. In doing so the manuscript describes new structure for SCC with one major and two minor Glc modifications. The manuscripts uses a range of methods to characterise the roles of the four proteins, including recombinant protein expression and genetic knockouts, coupled with carbohydrate structural analysis by MS and NMR.

We thank the reviewer for careful reading and evaluation of our manuscript.

Minor comments

Fig 2.A No units on the Y-axis

We thank the reviewer for highlighting this error. Units, nmol/min/mg⁻¹, have been added to the Y-axis of Fig. 2a.

Line 145-147 " Δ SccN membranes was dramatically reduced" Authors should quantify the reduction using the data i.e. 2-fold reduced etc

This sentence has been amended to read:

"Deletion of *sccN* reduces (~80%) the Glc content of SCC, whereas deletion of *sccP* has a relatively minor affect."

Major comment

Authors use linkage analysis of the SCC purified from various genetic knockouts to determine the roles of SccN, SccM and SccP amd Sccq. Assumptions are made based on the single knockouts that SccN and SccM are the major enzyme pair. Why is there no analysis of the SCC from the double knockout to confirm this?

The linkage analysis requires a large investment of time, labor and expense, so we limited ourselves to analysis of mutants that were most likely to yield new information. We found from linkage analysis of Δ *sccN* that the SccN product was required for 2,4-Rha branch. We included the SccM/SccQ double deletion mutant in the linkage analysis to confirm that neither of the two GT-C type transferases were required for the addition of the 4-Glc branch to 2-Rha (Fig. 4e). Thus, we concluded that there must be an additional GT-C type transferase that uses the SccN product for this modification. Furthermore, the single deletion mutants showed clearly that SccN and SccM were both essential for the glucosylation of the 2-OH of the 3-Rha moiety, and SccP and SccQ were both required for the glucosylation of the 4-OH of 3-Rha. It is very unlikely that we would learn anything new from the SccN/SccP, SccN/SccM or SccP/SccQ double knockouts.

To determine the roles of the glycosyltransferases, in addition to linkage analysis, we used NMR analysis of SCC variants purified from WT, $\Delta sccM$, $\Delta sccQ$, $\Delta sccN$ and $\Delta sccP$, and two double mutants, $\Delta sccN\Delta sccP$ and $\Delta sccM\Delta sccQ$. The results of NMR analysis provide strong support for the derived structure of SCC and the functions of SccN, SccM, SccP and SccQ enzymes in SCC biosynthesis.

Reviewer #3 (Remarks to the Author):

This manuscript builds on previous work by the same group (Zamakhaeva et al (2021)) and describes two novel glycosyltransferases found in the biosynthesis gene cluster for serotype c-specific carbohydrates (SCC) of *Streptococcus mutans*. They claim that these glycosyltransferases attach glucose moieties at different positions of the rhamopolysaccharide, working in tandem with two previously described glucosyl phosphoryl undecaprenol (Glc-P-Und) synthases, which synthesize Glc-P-Und in α -, or β -confirmation, respectively. Additionally, the authors claim that the more dominant α -glucosylation attached by the glycosyltransferase sccM is involved in biofilm formation, due to attachment of negatively charged GroP. Altogether, the authors characterize the SCCs of different mutants extensively via NMR and other analytical techniques. The manuscript seems to be overall valid but could profit from additional controls, text clarifications, and simpler description.

The authors appreciate the reviewer's kind words and careful review of our manuscript. We would like to correct one apparent misunderstanding, however. SccN and SccP had not been previously identified as Glc-P-Und synthases, nor had the anomeric configurations of the undecaprenol-linked intermediates been established, or their roles in glucosylation of polyrhamnose been identified. The information in this report is entirely novel.

The following points should be addressed:

1. Line 35: The authors frequently talk about "major" and "minor" Glc modifications. They however do not explain what is meant by that. If it is just describing the amount of Glc on the SCC, that should be stated.

By major, most abundant is intended. We have changed the sentence to read: This study reveals that SCC has one predominant and two more minor Glc modifications.

2. Line 234: The authors frequently mention 2,3-substituted Rha, however if I understand correctly, they mean that the Rha is substitute with Glc, but also with the next Rha in the backbone polymer, which makes it difficult to distinguish. – Could this be stated more clearly, to distinguish between backbone linkage and Glc linkage?

It is an important question, and we apologize if it is unclear. The only unambiguous way we could think of would be to write the chemical structure out as in: $\rightarrow 2)\text{-}\alpha\text{-L-Rhap-(1}\rightarrow 3)[\alpha\text{-D-Glcp-(1}\rightarrow 2)]\text{-}\alpha\text{-L-Rhap-(1}\rightarrow$. But, this gets to be

extraordinarily cumbersome when referring to multiple glycans. We feel that in each case we make it clear which linkage is involved in the polyrhamnose backbone and which constitute sites of attachment of the glucose branches. However, we have included precise structural descriptions corresponding to our abbreviated designations in a footnote to Supplementary Table 2.

3. Fig 3: Especially in the WT, but even in the absence of the dominant sccM glycosyltransferase (can it be called dominant?), β -Glc attachment mediated by sccQ at the C4-position is rather low, as shown in Fig. 3. Why is it so inactive. Could it have other functions than the one suggested in the manuscript?

SccQ does not compete with SccM for the substrate, β -Glc-P-Und, but only for the acceptor Rha residue. The MS data of the Glc-P-Unds isolated from the various strains implies the alpha and beta isomers are both present in roughly similar amounts in the WT strain. Thus, these data suggest that SccQ might have either lower expression or enzymatic activity in comparison to SccM. Direct enzymological analysis might shed some light on this question, but that is beyond the scope of this study.

4. Upon deletion of sccQ, an increase in 2,3-Rha would be expected (Fig 3, D) according to the model proposed in Fig 6. But the opposite is observed. How can this be explained?

In bacteria, lipid carrier, Und-P is maintained at low levels in the cytoplasmic membrane (approximately 10^5 Und-P molecules/cell, PMID: 19110475), and its availability forms a critical bottleneck for the synthesis of cell envelope components such as peptidoglycan and cell wall polysaccharides (SCC). Deletion of SccQ causes the SccP product, α -Glc-P-Und, to accumulate in the “dead-end” intermediate which sequesters Und-P limiting the available Und-P pool for other reactions including SccN/SccM-catalyzed attachment of Glc to the 2-OH of the 3-Rha. Consequently, reduced levels of Und-P also lead to cell shape defects in Δ sccQ because peptidoglycan synthesis is likely decreased also. We added this explanation in the Discussion section.

5. According to Figure 3E, less than half of all rhamnose units in the polymer are modified with Glc. Could something other than Glc be attached, like the D-ala modification of wall teichoic acid (WTA)?

Data from the literature estimates that ~ 50% of the rhamnose units in the polyrhamnose backbone are glucosylated, but our published data (Zamakhaeva et al, 2021) showed that WT SCC has a substantial portion of neutral polysaccharide with decreased Glc content. The non-reducing terminal Glc residues detected in the linkage analysis match the sum of the various branched rhamnose units pretty closely, suggesting that glucose can account for the branching component. In addition, rhamnopolysaccharides, similar to WTA,

contain a linkage unit which is likely composed of a short poly-Rha linker. The structure of this region is under investigation.

6. Fig3E: How can it be explained that deletion of *sccQ* reduces the amount of Glc in SCCs, but deletion of *sccP* has no effect? Is the model in Figure 6 incomplete or can *sccQ* use β -Glc-P-Und as additional substrate?

We do not think *SccQ* uses both Glc-P-Unds as substrates. Please see our explanation about the impact of the *SccQ* deletion on Und-P levels. Deletion of *SccP* does not limit the free Und-P pool. Because of this, attachment of α -Glc to position 2 of 3-rhamnose is not reduced in Δ *sccP* but is decreased in Δ *sccQ*.

7. The authors claim that no GroP is attached to the *sccQ* linked β -Glc modification of Rha. Deletion of *sccQ* however reduces the amount of phosphate in the SCCs (Fig. 3F). The observation that GroP is not transferred to *sccQ* linked β -Glc is based on SDS-Page of ANDS labeled SCCs which observes that deletion of *sccM* prevents the addition of charged GroP. However, is it possible that GroP is transferred from 2-linked α -Glc to 4-linked β -Glc? Or could the charge change due to low addition of β -Glc be too small to be visible in the SDS-Page? A Gro measurement as performed in the previous paper by Zamakhaeva et al (2021) should be used.

As this reviewer has pointed out, Δ *sccQ* has reduced attachment of α -Glc to 2,3-linked Rha (Fig. 3d), which we attribute to a pleiotropic effect of sequestration of Und-P into a 'dead-end' intermediate (α -Glc-P-Und). Accordingly, this mutant has a decreased number of Glc sites available for GroP modification. As the result, the Δ *sccQ* mutant has a decreased level of GroP in SCC. The observation that GroP is not transferred to *SccQ* linked β -Glc is based on NMR analysis demonstrating that the Glc attached to the 2-position of 3-Rha is the GroP acceptor site. A Gro measurement would not provide information about the attachment site for GroP on Rha.

8. Phages of *S. mutans* are proposed to bind the Glc modifications of SCCs. This could be tested with the new mutants.

Firstly, isolation of *S. mutans* phages is very rare (PMID: 26398909,34063251). Secondly, even if the *S. mutans* c serotype-specific phage is found, many bacterial strains possess various defense mechanisms against phage infection (restriction-modification systems, abortive infection, and CRISPR-Cas systems, etc), demonstrating no plaque formation. Lastly, the experiments with phages are beyond the scope of this study.

Minor:

1. An overview of these serotype specific rhamnopolysaccharides would help the reader. - It could be added to figure 1.

The overview of serotype specific *S. mutans* rhamnopolysaccharides was provided in Introduction. “Alternatively, serotype e has β -Glc attached to the corresponding hydroxyl group of Rha²⁹. Serotype f contains α -Glc or (Glc)₂ side-chain attached to the 3-position of 2-linked Rha and serotype k has an α -galactose side-chains attached to the corresponding position on Rha¹⁴”. We have no additional information to include. The mechanisms of biosynthesis of these rhamnopolysaccharides have not been investigated yet. Furthermore, the molecular structures of the polysaccharides require re-examination in light of our enhanced understanding of SCC structure and biosynthesis, and the development of improved analytical techniques.

2. Line 184: Is this the m/z for Glc-P-Und? - Could the name be added directly to the mass?

In the last paragraph prior to the determination of the anomeric configuration we state:

“In addition, a co-migrating compound, purified from the *S. mutans* membrane fraction by organic solvent extraction and preparative TLC, yielded a molecular ion, m/z =1007.7 by ESI-MS which is expected for Glc-P-Und (Supplementary Fig. 1d).”

3. Line 202-204: This deduction seems overall logical. - However, could NMR of purified Glc-P-Und confirm these findings?

It could, but it would require a much larger quantity of material and a much more extensive purification of notoriously unstable compounds. MS methods allow the determination of the anomeric configurations of sugar 1-phosphates at the picomole scale.

4. Line 216-217: Are we talking about the linkage unit GlcNAc? Could this be more specific? Could this be added to Fig 6?

The 4-N-acetylglucosaminitol is derived from the reducing end of the polysaccharide (and is the sugar involved in the phosphodiester linkage). We would prefer not to include this linkage in Figure 6 because, the linkage unit is not the subject of Figure 6 and including membrane (or cell wall) attachment would unnecessarily clutter the figure and imply some sort of spatial association of the glucosylations with the linkage (which we don't want to do) and linkage unit structure, which is still under investigation. The t-GlcNAc detected is currently hypothesized to be a capping sugar, used to terminate polyrhamnose elongation, and this is also still under investigation.

5. Line 217: What is the difference between the reducing end GlcNAc and non-reducing end GlcNAc?

These rhamnopolysaccharides are released from the cell wall by mild acid hydrolysis of the GlcNAc-phosphate in the linkage unit. This generates a polysaccharide with a free GlcNAc at the reducing end. This GlcNAc is in an open-ring form and is actually an aldehyde (since it is an aldose sugar in equilibrium with the 5-OH, to form a closed 6 member-ring) and so can be chemically reduced, for example by NaBH₄. The non-reducing end GlcNAc (t-GlcNAc) is a sugar with no attached substituents, i.e. it terminates a branch of the main polymer. It is attached at its -OH on carbon 1 to another sugar and so is not in an open-ring form. In this case, it theoretically terminates the main polymer, because all the branches are formed by t-Glc (i.e. unmodified glucose units, so they terminate their own branches).

6. Line 224: What could be the explanation of this observation? - You would usually expect that Glc contents are not reduced more than half if only one transferase is deleted. Often these glycosyltransferases can even complement for each other.

We have included the following information to explain the reduction in Glc levels in the $\Delta sccM$ and $\Delta sccQ$ SCCs:

“However, as is explained below, genetic deletion of GT-C type transferases renders the associated Und-P-linked intermediate into a ‘dead-end’ synthetic product. The Und-P in this ‘dead-end’ product cannot be re-cycled and consequently accumulates in the cell, reducing the cellular Und-P available for polysaccharide glucosylation (as well as peptidoglycan synthesis), resulting in pleiotropic effects on polyrhamnose synthesis”.

7. Line 231: Why is there GlcNAcitol found in the SCCs?

These SCCs were released from cell wall by mild acid, freeing the reducing end GlcNAc. They were then reduced with sodium borohydride to reduce the reducing end of the polysaccharide, forming GlcNAcitol. Please see our published study on the rhamnopolysaccharides isolated from *Streptococcus pyogenes* using mild acid treatment followed by sodium borohydrate reduction (PMID: 35105886).

8. Line 500-501: Is serotype e only glycosylated with β -Glc? - Or is that unknown?

It has been reported that serotype e is only glycosylated with β -Glc. However, it is probably worth re-evaluation in light of our recent studies.

9. Line 1031: Is it not sccM which attaches the "major form" of glycosylations?

That is correct.

10. Fig 3: nmol Pi per 50 nmol Rha or Pi/50 Rha.

Fig 3g has been corrected to read nmol of Pi per 50 nmol Rha (thank you).

11. Fig 6: Should the SCC not be drawn linked to the membrane via the GlcNac linker? I assume it is attached at this state.

Please see reply to minor comment 4.

Sincerely,

Natalia Korotkova